bs

# CCN measurements at the Princess Elisabeth Antarctica Research Station during three austral summers

Paul Herenz[1], Heike Wex[1], Alexander Mangold[2], Quentin Laffineur[2], Irina V. Gorodetskaya[3,4], Zoë L. Fleming[5], Marios Panagi[5], and Frank Stratmann[1]

[1]Leibniz Institute for Tropospheric Research, Leipzig, Germany
[2]Royal Meteorological Institute of Belgium, Brussels, Belgium
[3]Centre for Environmental and Marine Studies, Department of Physics, University of Aveiro, Aveiro, Portugal
[4]Department of Earth and Environmental Sciences, KU Leuven, Belgium
[5]National Centre for Atmospheric Science, Department of Chemistry, University of Leicester, Leicester, UK

*Correspondence to:* Heike Wex (wex@tropos.de)

**Abstract.**

For three austral summer seasons (2013-2016, each from December to February) aerosol particles arriving at the Belgian Antarctic research station Princess Elisabeth (PE), in Dronning Maud Land in East Antarctica were characterized. in terms of This included number concentrations of total aerosol particles ($N_{CN}$) and cloud condensation nuclei ($N_{CCN}$), the particle number size distribution (PNSD), the aerosol particle hygroscopicity and the influence of the air mass origin on $N_{CN}$ and $N_{CCN}$. In general $N_{CN}$ was found to range from 40 to $6700\,\mathrm{cm}^{-3}$ with a median of $333\,\mathrm{cm}^{-3}$, while $N_{CCN}$ was found to cover a range between less than 10 and $1300\,\mathrm{cm}^{-3}$ for supersaturations ($SS$) between 0.1 and 0.7 %. It is shown that the aerosol is Aitken mode dominated and is , being characterized by a significant amount of freshly, small, and therefore likely secondarily formed aerosol particles, with 94 % and 36 % of the aerosol particles are smaller than 90 nm and $\approx 35\,\mathrm{nm}$, respectively. Measurements of the basic meteorological parameters as well as the history of the air masses arriving at the measurement station indicate that the station is influenced by both, continental air masses originating from the Antarctic inland ice sheet (continental events - CEs) and marine air masses originating from the Southern Ocean and coastal areas around Antarctica (marine events - MEs) and continental air masses (continental events - CEs). CEs, which were defined as times when the air masses spent at least 90% of the time during the last 10 days over the Antarctic continent, occurred during 61% of the time during which measurements were done. CEs came along with rather constant $N_{CN}$ and $N_{CCN}$ values, which we denote to be Antarctic continental background concentrations. MEs however cause large fluctuations in $N_{CN}$ and $N_{CCN}$ with low concentrations likely caused by scavenging due to precipitation and high concentrations likely originating from new particle formation (NPF) based on marine precursors. The application of HYSPLIT back trajectories in form of the potential source contribution function (PSCF) analysis indicate, that the region of the Southern Ocean is a potential source of Aitken mode particles. On the basis of PNSDs, together with $N_{CCN}$ measured at a $SS$ of 0.1 %, median values for the critical diameter for cloud droplet activation and the aerosol particle hygroscopicity parameter $\kappa$ were determined to be 110 nm and 1, respectively. For particles larger than $\approx 110\,\mathrm{nm}$ (CCN measured at $SS$ of 0.1 %) the Southern Ocean together with parts of the Antarctic

ice shelf regions were found to be potential source regions. While the former may contribute sea spray particles directly, contribution of the latter may be , most likely due to the emission of sea salt aerosol particles, released from snow particles from surface snow layersby sublimation, e.g., during periods of high wind speed, leading to drifting or blowing snow. On the basis of the PNSDs and $N_{\mathrm{CCN}}$, the critical diameter for cloud droplet activation and the aerosol particle hygroscopicity parameter $\kappa$ were determined to be 110 nm and 1, respectively, for a $SS$ of 0.1 %. The region of the Antarctic inland plateau however was not found to feature a significant source region for CN and CCN measured at the PE station in austral summer.

## 1   Introduction

Aerosol particles can be emitted into the atmosphere either directly e.g., by mechanical processes or combustion, or indirectly, due to nucleation from the gas-phase. Under specific conditions, aerosol particles can act as cloud condensation nuclei (CCN) and form cloud droplets. Whether a particle forms a cloud droplet depends on its size and the chemical composition and the size of the particle as well as the surrounding supersaturation (SS). Aerosol particles can influence the climate either directly, by scattering or absorption of solar radiation, or indirectly due to their impact on cloud formation and cloud properties such as e.g., the cloud albedo (Twomey, 1974) or on the lifetime of clouds (Albrecht, 1989; Rosenfeld et al., 2008). The direct effect of aerosol particles is relatively well understood. In contrast to this, the manifold indirect aerosol effects are less understood. The influence of indirect aerosol effects on the global climate and the radiative forcing still features a low confidence level and large uncertainties (IPCC, 2013). The climate impact of aerosol particles on climate and global radiative forcing is mainly determined by their physical and chemical properties. Investigations of these properties of aerosol particles in general and of CCN in particular by means of in situ measurements at various different sites and conditions are a necessary to lower these uncertainties. The Antarctic region is particularly interesting for aerosol particle and CCN in situ studies for two reasons. Firstly, Antarctica is located far from anthropogenic activities and is one of the most pristine areas on the globe (Hamilton et al., 2014). Thus, it is a favorable environment for studying natural aerosol particle background conditions and processes that prevailed in a preindustrial atmosphere. A more accurate knowledge about preindustrial aerosol processes, conditions and properties, including aerosol-cloud interactions, is important for a reduction of uncertainties of model estimates concerning radiative forcing (Hamilton et al., 2014; Carslaw et al., 2013). Secondly, similar to the Arctic, the Antarctic region is extremely sensitive to climate change. Jacka and Budd (1998) analyzed surface temperature data of 16 stations on the Antarctic continent and 22 stations on Southern Ocean islands and found warming rates of 0.9-1.2 °C and 0.7-1 °C, respectively. In particular in the West Antarctic and the Antarctic Peninsula the warming is several times higher than in other regions (Jacka and Budd, 1998; Vaughan et al., 2003; Kravchenko et al., 2011; IPCC, 2013). The Antarctic sea ice as well as the inland ice sheet are potentially subject to change in such a changing environment. However, at the moment, the Southern Hemisphere has not shown a decrease in sea ice extent. Cavalieri and Parkinson (2008) and Parkinson and Cavalieri (2012) even found an increasinga positive trend in the annual maximum Southern Ocean sea ice extend. Both, the sea ice area and the open water area have the potential to emit aerosol particles into the atmosphere. Sea ice is a potential source for sea salt aerosol particles (Huang and Jaeglé, 2017; Yang et al., 2008; Wagenbach et al., 1998) and nitrogen (Dall'Osto et al., 2017) and open sea water may emit sea spray aerosol

and precursors for new particle formation (NPF) (Liss and Lovelock, 2008; Modini et al., 2015). Therefore, variations in sea ice coverage will likely lead to changes in the nature of aerosol particle sources. The mass balance of the Antarctic ice sheet also shows quite unexpected trends. There are opposing trends in the ice sheet mass balance across Antarctica. Velicogna and Wahr (2006) and Shepherd et al. (2012, 2018) found that the ice sheets of West Antarctica and the Antarctic Peninsula had

lost mass, whereas the East Antarctic ice sheet had gained mass. The gain of ice mass in East Antarctica is also confirmed by Martin-Español et al. (2017), however they found it to be smaller than losses in West Antarctica. As precipitation, which in addition to moisutre amounts is also strongly linked to the abundance of CCN and ice nucleating particles, is the only source of mass gain to the Antarctic ice sheet, it is necessary to study the properties of these aerosol particles as well as their impact on cloud formation and precipitation, their sources, sinks and pathways in the changing environment of Antarctica.

Although Antarctica is a harsh environment where access for field work is difficult, various aerosol particle studies have been conducted at different Antarctic research stations during the last decades. A wide range of topics has already been investigated, including new particle formation (Koponen et al., 2003; Asmi et al., 2010; Kyrö et al., 2013; Fiebig et al., 2014; Weller et al., 2015), seasonal cycles of number and mass concentrations as well as size distributions (Koponen et al., 2003; Kim et al., 2017; Fiebig et al., 2014), chemical composition (Wagenbach et al., 1988; Teinila et al., 2000), hygroscopicity (Asmi et al., 2010;

Kim et al., 2017; O'Shea et al., 2017) and optical properties of aerosol particles (Fiebig et al., 2014).

In general, there is a yearly trend in particle number concentrations, with maximum values in austral summer (Kim et al., 2017; Fiebig et al., 2014). Fiebig et al. (2014) conclude that these cycles are common across the Antarctic Plateau (including the Troll research station 235 km from the Antarctic coast but still 2000 km away from the South Pole), with free tropospheric air masses contributing to air detected at ground. The highest particle concentrations found in austral summer are frequently

reported to be due to NPF events (Asmi et al., 2010; Koponen et al., 2003; Kyrö et al., 2013; Weller et al., 2015; Kim et al., 2017). Particles formed during NPF events are likely related to sulfate and ammonia containing compounds that were found in the particulate phase in the submicron size range (Teinila et al., 2000; Wagenbach et al., 1988; Schmale et al., 2013). Precursor gases for NPF can originate from the Southern Ocean (e.g., DMS (dimethylsulfid), Weller et al., 2015; Schmale et al., 2013) and possibly also from e.g., cyanobacteria in freshwater melt ponds (Kyrö et al., 2013), microbiota from sea ice and the sea ice

influenced ocean (Dall'Osto et al., 2017) or the decomposition of excreta from fur seals, seabirds and penguins (Legrand et al., 1998; Schmale et al., 2013).

Newly formed particles were sometimes reported to grow to CCN size ranges at the Aboa research station, e.g., in Kyrö et al. (2013) and Koponen et al. (2003) (for the latter only in marine air masses), while Weller et al. (2015) report a maximum size of only 25 nm for particles grown from new particle formation for observations at the Neumayer research station. This difference

was related to a difference in ground cover at the respective measurement sites, which was ice covered around Neumayer but featured melt ponds around Aboa (Weller et al., 2015).

Sodium chloride, supposedly from sea spray, was found for larger particles (well above 100 nm) at Aboa, while the majority of particles was smaller than 100 nm (Teinila et al., 2000). A case of exceptionally high particle hygroscopicity was connected to air masses originating from a region with sea ice and open water at the coastal Antarctic research station Halley (O'Shea et al.,

2017). Asmi et al. (2010) assumed that particles and nucleating and condensing vapors from the Southern Ocean contribute

to particles observed at Aboa, and observed hygroscopic growth factors for particles of 25, 50 and 90 nm that were similar to those they reported for ammonium sulfate.

Furthermore, some studies have reported on Antarctic CCN properties, however the locations they cover are limited to the Antarctic Peninsula (DeFelice, 1996; DeFelice et al., 1997; Kim et al., 2017) or the area of the Weddell Sea on the Brunt Ice Shelf (O'Shea et al., 2017). Both locations are part of West Antarctica and especially the Antarctic Peninsula is mainly influenced by marine air masses that directly originate from the Southern Ocean. To create achieve a more detailed picture of Antarctic CCN, further measurements that can be used to characterize CCN in the eastern and especially in the central part (Antarctic inland plateau) of Antarctica, are needed.

To gain further knowledge about aerosol particle and particularly CCN properties in East Antarctica, we conducted measurements at the Belgian Antarctic research station Princess Elisabeth (PE), in Dronning Maud Land. For three austral summer seasons (2013-2016, always from December to February) a Condensation Particle Counter (CPC), a Cloud Condensation Nucleus counter (CCNc) and a Laser Aerosol Spectrometer (LAS) were used to measure simultaneously aerosol particle and CCN properties inside the East Antarctic boundary layer. In addition, the present study introduces meteorological data, collected by an automatic weather station and precipitation rates derived from a vertically pointing precipitation radar, as well as the history of the measured air masses, calculated by means of the Numerical Atmospheric-dispersion Modelling Environment (NAME) and the NOAA HYSPLIT trajectory model. This data set has enabled the study of the variability of the condensation Nuclei (CN) and CCN number concentrations, to identify their sources, sinks and transport pathways and to analyze the particle hygroscopicity during austral summer in East Antarctica. The special location of the PE station in the escarpment zone with katabatic winds coming from the Antarctic inland ice sheet, further allows an insight into the state of aerosol particle and CCN properties of continental Antarctica.

## 2 Experimental procedure and methods

### 2.1 Measuring site and meteorology

The measurements presented in this study were all performed at the Belgian Antarctic research station Princess Elisabeth ( PE station (Figure 1a), in Dronning Maud Land, East Antarctica (71.95° S, 23.35° E, 1390 m asl., around 200 km inland from the Antarctic coast). The PE station is located upon the granite ridge of Utsteinen Nunatak in the Dronning Maud Land region of East Antarctica and lies north of the Sør Rondane Mountain Range, that has peaks up to an altitude of 3300 m asl. This area is located in the escarpment zone between the Antarctic inland plateau and the coast which can be seen in the topographic map of Antarctica in Figure 1b. A more detailed description of the conditions at the measurement station and its near surroundings is given by Pattyn et al. (2010) and Gorodetskaya et al. (2013). The PE station is designed as a zero-emission station with power production mainly based on wind and solar energy (see www.antarcticstation.org). This reduces local emissions which makes the PE station an excellent base for conducting in situ aerosol particle measurements. Nevertheless, general station activities, traffic by skidoos or bulldozers, and irregular diesel generator operation times cause contamination which is however removed from discarded in the final data (see section 2.2). The station is inhabited from November through the end of February.

During the other months the station and most of its scientific instruments are operated under remote control. As the Cloud Condensation Nuclei counter used for this study needs an operator on site, we mainly present data collected from December to February during three subsequent austral summers (2013-2016).

The basic meteorological parameters (near-surface air temperature, pressure, relative humidity, wind speed and wind direction) were measured by means of an automatic weather station (AWS, Gorodetskaya et al., 2013; Souverijns et al., 2018) which was located 300 m east of PE. Snowfall rates (S) were derived from Metek's micro-rain radar (MRR-2) effective reflectivities (Z) applying a Z-S relationship derived specifically for the PE location (Souverijns et al., 2017). The radar is a vertically pointing 24-GHz precipitation radar operating at PE since January 2010 (Gorodetskaya et al., 2015). Radar Z were estimated from the radar's raw Doppler spectra using the algorithm of Maahn and Kollias (2012) specifically designed for MRR snowfall applications. S at 300m agl level were estimated using a mean Z-S relationship derived by Souverijns et al (2017) based on snowfall microphysical measurements at PE. More details about the AWS, the precipitation radar and the estimation of precipitation can be found in Gorodetskaya et al. (2013), Gorodetskaya et al. (2015) and Souverijns et al. (2017), respectively and on the AEROCLOUD website (http://www.aerocloud.be). Generally, the meteorological situation at PE is characterized by either synoptic regimes, which usually correspond to strong easterly winds with sometimes a slight northerly component, or a katabatic regime, that is mostly associated with relatively weak south-southeasterly winds (Gorodetskaya et al., 2013; Souverijns et al., 2018).

The two different meteorological situations can be identified based on wind speed and wind direction, which are both depicted in the form of a wind rose in Figure 1c (exemplarily based on measurements between December 18, 2015 to February 20, 2016 in the third season). The more frequent easterly winds clearly correspond to higher wind speeds, mainly over 5 m/s, whereas the less frequently occurring southerly winds are usually below 5 m/s. Additional meteorological parameters for each year are shown in Figure 2 as a time series of hourly (gray lines) and daily averaged (red lines) values. Respective seasonal mean values together with the standard deviation, minimum and maximum for the period from December 1 to February 20 are shown in Table 1. The mean values as well as the fluctuation in the meteorological parameters show no large differences between the three measurement periods. Due to the shielding effect of the Sør Rondane Mountains to the South, blocking the katabatic winds from the inland plateau, the climate at PE is relatively mild for for the Antarctic escarpment zone (Gorodetskaya et al., 2013).

## 2.2 Instrumentation and data processing

The total particle number concentration ($N_{CN}$) was measured by a Condensation Particle Counter (CPC, TSI model 3776), which has a lower cut off at 3 nm and was operated at a total flow rate of 1.5 l/m. The CPC was first installed for continuous operation in November 2012. Due to several power outages in austral winter, data coverage of the winter months was not equal between the years and the CPC was restarted in the respective austral summers. The last data for this study was measured in May 2016. The inlet tubing for the CPC consisted of a 1 m long vertical 0.5 inch stainless steel tubing (not heated) installed through the roof of the measurement container. Inside, 0.7 m of a 3/8 inch (0.19 inch inner diameter) conductive flexible tubing made the connection to the CPC in a smooth bend from the ceiling of the container down to the CPC. with only a smooth bend

just before the inlet at the front of the CPC. On the roof of the container, $0.15\,\mathrm{m}$ of the same flexible tubing was connected to the stainless steel tubing in order to serve as inlet without a size cut-off. With this kind of inlet, there were never issues with inlets clogged by snow during storms, in particular important during the non-inhabited winter periods. Clogging of inlets caused by riming never happened due to the extreme dryness at the measurement site. The CPC was operated with a $4\,\mathrm{l}$ butanol reservoir bottle. Consumption of butanol was between 3 to $3.5\,\mathrm{l}$ for a complete year of measurements. Each austral summer the CPC was checked on leaks and the butanol was exchanged. The procedure to assure non-contaminated data is described further below.

In parallel to $N_{\mathrm{CN}}$ the particle number size distribution ($PNSD$) was measured by means of a Laser Aerosol Spectrometer (LAS, TSI model 3340) in the size range from $90\,\mathrm{nm}$ up to $6.8\,\mu\mathrm{m}$ (99 log-distributed channels). The inlet setup and tubing for the LAS is similar to the one for the CPC and installed directly next to that one. However, inside, first $0.5\,\mathrm{m}$ of a 1/8 inch (inner diameter) and then $0.2\,\mathrm{m}$ of a 1/16 inch (inner diameter) conductive flexible tubing made the connection (no bend) to the measurement chamber of the LAS. The LAS was operated with a sample flow rate of $0.07\,\mathrm{l/min}$ with a sheath flow of $0.6\,\mathrm{l/min}$. While no aerosol drying was installed in front of the LAS, ambient humidities and temperatures together with temperatures inside the container and the LAS were such that it can safely be assumed that relative humidities in the aerosol sampled for size distribution measurements were below 20%. The LAS was maintained and re-calibrated in spring 2015 by TSI Inc. In October 2015, before shipment to Antarctica, the LAS was compared to an SMPS system (DMA type Medium Hauke; CPC, TSI model 3010) at the cloud laboratory of the Leibniz Institute for Tropospheric Research (TROPOS). 16 selected sizes (80, 90, 100, 110, 120, 130, 140, 150, 175, 200, 250, 300, 350, 400, 500, 600, 700, 800 nm) of ammonium sulfate particles and 4 sizes (100, 500, 700, 800 nm) of PSL™ standard solutions were used to validate the counts of the LAS. At $500\,\mathrm{nm}$, both ammonium sulfate and PSL particles resulted in similar signals, however, at $100\,\mathrm{nm}$ signals for PSL appeared in a broad range of channels, so here only ammonium sulfate particles were used for the validation. Necessary corrections were high in the two LAS channels below $100\,\mathrm{nm}$ (around +70 %), distinct + 10 % in the two channels around 100 nm (+ 10 %) and low in the other size ranges up to $800\,\mathrm{nm}$ (between 1 to 5 %, negative and positive corrections). These corrections were applied to the LAS data set used for this study. In this study, we continuously use hourly averaged $N_{\mathrm{CN}}$ values and $PNSD$s.

The number concentration of cloud condensation nuclei ($N_{\mathrm{CCN}}$) was measured using a Cloud Condensation Nuclei counter (CCNc, Droplet Measurement Technologies (DMT), Boulder, USA). The CCNc is a continuous-flow thermal-gradient diffusion chamber which is described in detail by Roberts and Nenes (2005). The inlet tubing for the CCNc consisted of a $2.2\,\mathrm{m}$ long vertical conductive flexible tubing (similar to the one used for the CPC and LAS) with only a smooth bend just before the inlet of the CCNc. The inlet outside was directly next to the inlets of the CPC and LAS. The CCNc was operated as recommended by Gysel and Stratmann (2013) for polydisperse CCN measurements. The CCNc was operated at a constant total flow rate of $0.5\,\mathrm{l/m}$ and at 5 different supersaturations (SS; 0.1, 0.2, 0.3, 0.5 and 0.7 %), each for 12 minutes per hour. To ensure stable column temperatures, the first $5\,\mathrm{min}$ and the last $30\,\mathrm{sec}$ at each $SS$ setting were excluded from the data analysis. The remaining data points were averaged, so that the result is one $N_{\mathrm{CCN}}$ value per $SS$ per hour. For consistency checks between $N_{\mathrm{CN}}$ and $N_{\mathrm{CCN}}$, additional measurements at a supersaturation of also 1 % $SS$ was adjusted at times (but data at that SS were not included in the analysis presented in here) were made a few times during each season. Respective values for $N_{\mathrm{CCN}}/N_{\mathrm{CN}}$

generally were between 0.8 and 0.9. As we will discuss later, the aerosol at PE station is strongly dominated by particles in the nucleation- and Aitken-mode size range, and at 1% $SS$, not all particles were activated during times when the supersaturation was set to 1% (for example, activation of particles will occur down to 36 nm and 24 nm for an hygroscopicity parameter $\kappa$ of 0.3 and 1, respectively). Hence this consistency check could not be applied here. But prior to each of the three measurement

periods in Antarctica a $SS$ calibration of the CCNc was done at the cloud laboratory of TROPOS. These calibrations were performed with size selected ammonium sulfate particles and for pressure conditions relevant for the PE station (approximately 820 hPa), based on recommendations given by Gysel and Stratmann (2013) and Rose et al. (2008). Besides for calibration curves for the CCNc, also $N_{CCN}/N_{CN}$ were derived for particles of different sizes between 120 nm and 200 nm at $SS$ between 0.2% and 0.7%. On average $N_{CCN}/N_{CN}$ were 1.01, 0.99 and 0.96 for the three different years. All values ($N_{CN}$, $N_{CCN}$ and

$PNSD$) are presented with respect to standard conditions, i.e., a pressure of 1013.25 hPa and a temperature of 293.15 K.

In addition, also data from an aethalometer (Magee Sci. AE31, 7-wavelength aethalometer) was used. The aethalometer was operated with an inlet flow of 5.5 l/min and, similar as for the other instruments, the tubing through which it was fed was 2 m of flexible conductive tubing, including the inlet on the roof of the measurement container. The measurement interval was set to 60 min. Aethalometer data were analyzed following the guidelines in WMO (2016).

The container for the aerosol measurements is located 60 m south of the PE main station (Figure 1a). It was most often exposed to non-contaminated air due to the facts Given that the PE station is designed as a zero-emission station and that the daily activities are concentrated in the W-NW sector while the main wind directions are from south to east and the distribution of the wind direction (Figure 1c). , the container was most often exposed to non-contaminated air. The container is well-insulated and equipped with a small heater. Heating was hardly necessary in austral summer (due to 24 hour sun light). However, there

is no air conditioning system (due to energy demand; remote control during austral winter; necessary filter systems and no exchange possible of them during austral winter). Therefore, in austral summer, the temperature inside the container varied between $\approx$ 10 and 40 °C. This range exceeds the recommended operating temperature range of the CPC and the LAS of 10 to 35 °C and 10 to 30 °C, respectively, as well as the temperature range for which the CCNc was calibrated, which is 20 to 30 °C. Therefore, $N_{CN}$, $N_{CCN}$ and $PNSD$s measured during time periods in which the temperature inside the measurement

container was outside of the operating temperature ranges were excluded from the analysis presented here.

Further, as mentioned in section 2.1, the data still contained values caused were still partly influenced by emissions from the activities at the station. In order to identify hourly intervals with contamination, the following data sets were examined: i) the minute-by-minute $N_{CN}$ -CPC data; ii) simultaneously measured hourly data for the mass concentration of light-absorbing aerosol measured with the aethalometer (Magee Sci. AE31, 7-wavelength aethalometer; set up in the same measurement con-

tainer); iii) wind speed and wind direction measured by the AWS. As indicators for contamination abrupt peaks, outliers, and strong variations between higher and lower minute-by-minute $N_{CN}$ -CPC values and/or distinctly higher mass concentrations of light-absorbing aerosol ($>50$ ng/m$^3$) were used. Because the PE station was designed as zero-emission station, there was no relationship between wind speed or wind direction with elevated values for $N_{CN}$ or light-absorbing aerosol. However, each hourly interval with wind speed $<$ 3 m/s and/or wind direction outside the sector 20° to 225° was examined again for conspic-

uous signals in its variation in time.

The hygroscopicity of the aerosol particles was determined by applying the $\kappa$-Köhler-theory (Petters and Kreidenweis, 2007). The inferred hygroscopicity parameter $\kappa$ represents the average particle composition chemistry. To infer $\kappa$, first the critical diameter ($d_{crit}$) needs to be determined, based on the measured $N_{\text{CCN}}$ and $PNSD$. $d_{crit}$ This is the diameter at which particles are just large enough to be activated to a droplet when exposed to a certain supersaturation (SS). For a pair of simultaneously measured $PNSD$ and $N_{\text{CCN}}$, $d_{crit}$ is obtained by calculating the cumulative particle number concentration from that $PNSD$, from the largest diameter on downward, and it is the diameter at which this cumulative concentrations is equal to $N_{\text{CCN}}$. Using the assumption that the surface tension is equal to that of pure water, $d_{crit}$ and the SS are used to derive $\kappa$ values of the Antarctic aerosol particles. This approach, however, does assume that all particles of roughly the size of $d_{crit}$ have the same chemical composition, i.e., are internally mixed. Therefore the derived $\kappa$ values will only give a rough information on the chemical composition of the examined aerosol, which, however, still can be useful in interpreting the origin of the observed aerosol particles. A detailed description of this method, including the application of a Monte Carlo simulation to precisely determine uncertainties in $d_{crit}$ and $\kappa$, is presented by Herenz et al. (2017). This procedure of inferring $\kappa$ values could only be done for $N_{\text{CCN}}$ measurements at a $SS$ of 0.1 %, as $d_{crit}$ for larger $SS$ is below the lower size limit of the $PNSD$s of 90 nm.

## 2.3 Identification of air mass origins and potential source regions

To analyze the influence of the air mass origin on $N_{\text{CN}}$ and $N_{\text{CCN}}$ measured at the PE station, we applied two different models to obtain information on the air mass history. The first one is the Numerical Atmospheric-dispersion Modelling Environment (NAME), which was used to perform a simple residence time analysis (Fleming et al., 2012). The second one is the Potential Source Contribution Function (PSCF), a more advanced type of residence time analysis that results in a probability field which represents the probability of a specific location to contribute to high measured receptor concentrations (Fleming et al., 2012). As will be described in more detail below, the two models are based on different sets of back-trajectories, i.e., they were used in the framework in which they have been tested and applied in the past.

### The NAME dispersion model

The NAME atmospheric dispersion model (Jones et al., 2007) is a Lagrangian particle-trajectory model, that is operated by the UK Meteorological Office. For this study 10000 abstract particles per hour were released at 10 m above the location of the PE station. On the basis of the Meteorological Office Unified Model (UM) meteorological field data, 10-day back trajectories for these particles were calculated. Summing up the concentration of these particles at time steps of 15 minutes back in time (in total 960 time steps) results in a footprint that shows the history of the air masses during the last 10 days. For this procedure only particles that are located within the surface layer (i.e., 0-100 m above ground) are taken into account. An example footprint of the first of December 2013 (midnight) is shown in Figure 3. Footprints were derived every three hours, resulting in a total number of 2019 NAME footprints used in this study. To further analyze the impact of different surface properties on the measured aerosol particle properties, the area around Antarctica was divided into the following 5 different regions (see also Figure 4): the Antarctic escarpment zone and inland plateau (continental area at or above 200 m above sea surface level (asl)), Southern Ocean, South America, Africa and ReactiveProductive Zone. In marine regions at lower latitudes, sea spray particles

generally contribute only small fractions to total particle and CCN number concentrations (Wex et al., 2016; Quinn et al., 2017). However, at latitudes above 40° (both N and S), this fraction increases, and, due to prevailing high wind speeds, the Southern Ocean may contribute sea spray particles (in a mode with sizes from roughly 100 nm well up into the supermicron size range) which may make up 20% to 30% of all particles, at least above the ocean (Quinn et al., 2017). But the Southern Ocean also is a source for precursor gases for NPF such as DMS and ammonia (Schmale et al., 2013). These precursors may originate in phytoplankton blooms correlated to increased chlorophyll concentrations and have been described to influence CCN over the Southern Ocean (Vallina et al., 2006; Meskhidze and Nenes, 2006). The ReactiveProductive Zone includes the following regions that are known to have the potential to emit either primary particles (i.e., particles from sea spray in this case) or precursors for secondarily formed particles (i.e., for NPF):

 – The Antarctic continental area below 200 m asl and 8 islands in the Southern Ocean (South Georgia, South Sandwich, Falkland, South Orkney, Prince Edward, Crozet, Kerguelen, Heard and McDonald Island). These regions are included because they are habitats for numerous different types of penguins and birds. Bird guano (Schmale et al., 2013) or (Croft et al., 2016) and in this special case penguin guano (Legrand et al., 1998), acts as a source of ammonia and may contribute to the formation of new particles in coastal Antarctic areas. Also, cyanobacteria from freshwater melt ponds have been described to contribute precursor gases for NPF and particle growth (Kyrö et al., 2013).

 – The permanently and seasonally covered sea ice areas. These are known to have the potential to act as source of organic nitrogen that contributes to secondarily formed aerosol particles (Dall'Osto et al., 2017) or to emit primary sea salt particles (Huang and Jaeglé, 2017; Yang et al., 2008; Wagenbach et al., 1998).

 – The marine area up to 200 km from the coasts of the islands and continents (for Antarctic, continent plus ice shelves). Sea spray production may occur in this region (Quinn et al., 2017). Also, these areas are included due to an enhanced chlorophyll concentration in the coastal areas of the Southern Ocean. As said before, chlorophyll can be used as a proxy for DMS (Vallina et al., 2006), which in turn plays a role in new particle formation (Liss and Lovelock, 2008).

The proportional residence time that the air masses spent over the 5 different regions during the last 10 days was determined in order to assess to what extent these regions influence the aerosol particle and CCN properties. This type of a residence time analysis was already used for an Antarctic site by O'Shea et al. (2017). The comparably coarse division into the different regions used in this study was thought to yield a general idea on the possible origin of particles or particle precursors. A more detailed investigation of, for example, the variability of the ice cover or the existence of phytoplankton blooms in the examined regions is beyond the scope of our study.

**Potential Source Contribution Function**

The Potential Source Contribution Function (PSCF) is a receptor modeling method that originally was developed by Ashbaugh et al. (1985) and was applied in a number of high latitude studies before e.g., in Dall'Osto et al. (2017) for the Antarctic and in Yli-Tuomi et al. (2003) for the Arctic. The PSCF model is based on air mass back trajectories, and it is commonly used to

identify regions that have the potential to contribute to high values of measured concentrations at a receptor site. In this study we apply the PSCF on $N_{CN}$ and $N_{CCN}$.

The NOAA HYSPLIT trajectory model (Stein et al., 2015) was used to calculate hourly resolved 10-day back trajectories based on 1x1° GDAS (Global Data Assimilation System) meteorological data. To account for uncertainties in back trajectory

analysis, every hour a set of 15 back trajectories was calculated, which is composed of 5 different plane locations (one exactly at the measurement station and 4 in close proximity around it) at three altitudes (100 m, 200 m and 300 m above the surface level). In total this results in a set of 88152 back trajectories that were used for the PSCF analysis (note that a few trajectories were excluded from the analysis as they could not be properly calculated due to problems in the input data). Each back trajectory consists of trajectory segment endpoints, which represent the central geographical position of the air parcel at a particular time.

To calculate the PSCF the whole region that is covered by these trajectory segment endpoints is divided into an array of 5x5° grid cells (i,j). The assumption is that aerosol particles that are emitted in such a cell are incorporated into the air parcel and transported to the receptor cite. The PSCF can be calculated as follows:

$$PSCF_{i,j} = \frac{m_{i,j}}{n_{i,j}}, \tag{1}$$

where $n_{i,j}$ is the total number of trajectory segment endpoints that fall into a cell and $m_{i,j}$ is the number of trajectory segment

endpoints that fall into that cell and fulfill a given criterion, where this criterion typically is the exceedance of a certain threshold. In this study we used the 75 % percentile of either $N_{CN}$ or $N_{CCN}$ as that threshold. According to Hopke (2016): "Cells containing emission sources would be identified with conditional probabilities close to 1 if trajectories that have crossed the cells effectively transport the emitted contaminant to the receptor site. The PSCF model thus provides a means to map the source potentials of geographical areas. It does not apportion the contribution of the identified source area to the measured

receptor data." As it is probable that small values of $n_{i,j}$ would lead to uncertain and high PSCF values it is necessary to apply a weighting function. For this study a discrete weighting function based on $\log(n+1)$, which is a measure of the back trajectory density, was applied (Waked et al., 2014):

$$W = \begin{cases} 1.00 & \text{for } n_{i,j} \geq 0.85 \cdot \max(\log(n+1)) \\ 0.725 & \text{for } 0.6 \cdot \max(\log(n+1)) > n_{i,j} \geq 0.85 \cdot \max(\log(n+1)) \\ 0.35 & \text{for } 0.35 \cdot \max(\log(n+1)) > n_{i,j} \geq 0.6 \cdot \max(\log(n+1)) \\ 0.1 & \text{for } 0.35 \cdot \max(\log(n+1)) > n_{i,j} \end{cases} \tag{2}$$

The measured concentration of total particles and CCN is also affected by losses that occur along the path of the air parcel

between the source and the receptor site. As precipitation, which is known to be one of the major sinks for aerosol particles, in particular for CCN, is an output parameter of the calculated NOAA HYSPLIT back trajectories, it can be taken into account. Hence, we run the PSCF model with a precipitation filter. Back trajectories were cut off, and not considered for the PSCF analysis, as soon as a trajectory segment endpoint shows a precipitation of 0.1 mm/h and the total precipitation (sum of

precipitation of 240 trajectory segment endpoint) of this back trajectory exceeds a value of $5 \, \mathrm{mm/240h}$. The second criterion was added as it seemed not to be reasonable to discard a trajectory only because of showing a low precipitation of some $\mathrm{mm/h}$ at some trajectory segment endpoints. While the precipitation filter criteria described here were particularly contrived for our study, we used the weighting function given in Waked et al. (2014), as already stated above. Note, the filter criteria as well as the criteria of the weighting function are empirical.

## 3 Results and discussion

### 3.1 Total Particle and CCN number concentrations and regional analysis of the NAME model footprints

This section presents the measured $N_{\mathrm{CN}}$, $N_{\mathrm{CCN}}$ and $PNSD$ as well as the proportional residence time of the air masses over the regions introduced in Section 2.3. Time series are given for the three austral summer seasons of 2013/2014, 2014/2015 and 2015/2016 in Figure 6, 7 and 8, respectively.

Measurements of $N_{\mathrm{CN}}$ throughout the whole year were performed between 2012 and 2016. Their visualization in Figure 5 shows a clear seasonal cycle with the lowest monthly median values during the austral winter and a maximum during late austral summer. The monthly $10\,\%$ and $90\,\%$ percentiles also indicate the highest variability of $N_{\mathrm{CN}}$ during February. Several studies at different Antarctic sites found that the physical and chemical aerosol particle properties are subject to a similar seasonality e.g., Hara et al. (2011), Weller et al. (2011), Virkkula et al. (2009) and Kim et al. (2017). Just like $N_{\mathrm{CN}}$, also $N_{\mathrm{CCN}}$ follows a seasonal cycle with a minimum in austral winter and a maximum in austral summer (Kim et al., 2017). Hence, our measurements during austral summer capture the season in which the aerosol production in Antarctica and the surrounding source regions is most active.

We found $N_{\mathrm{CN}}$ (black dots in panel C of Figure 6, 7 and 8) to cover a range between 40 and $6700 \, \mathrm{cm^{-3}}$ (on the base of hourly averaged values) with a median value of $333 \, \mathrm{cm^{-3}}$. Our measured $N_{\mathrm{CCN}}$ (bluish dots in panel C of Figure 6, 7 and 8) cover a range between less than $10 \, \mathrm{cm^{-3}}$ at SS=$0.1\,\%$ to $1300 \, \mathrm{cm^{-3}}$ for the highest largest $SS$ of $0.7\,\%$. The integration of the $PNSD$ over the whole size range ($N_{\mathrm{CN>90nm}}$, red dots in panel C of Figure 6, 7 and 8) shows the aerosol particle number concentration in the size range between 90 nm and $6.8\,\mu\mathrm{m}$. $N_{\mathrm{CN>90nm}}$ has a median value of $20 \, \mathrm{cm^{-3}}$. The median, 10 and $90\,\%$ percentile values for $N_{\mathrm{CN}}$, $N_{\mathrm{CN>90nm}}$ and $N_{\mathrm{CCN}}$ at all measured $SS$ are summarized in the first column of Table 2. O'Shea et al. (2017) and Kim et al. (2017) both also report $N_{\mathrm{CCN}}$ determined during austral summer, however, at coastal Antarctic locations. O'Shea et al. (2017) show $N_{\mathrm{CCN}}$ of approximately $20 \, \mathrm{cm^{-3}}$, $120 \, \mathrm{cm^{-3}}$ and $250 \, \mathrm{cm^{-3}}$ on average at $0.08\,\%$, $0.2\,\%$ and $0.53\,\%$ $SS$, respectively, and just under $200 \, \mathrm{cm^{-3}}$ at $0.4\,\%$ $SS$ are given in Kim et al. (2017). These $N_{\mathrm{CCN}}$ are roughly $50\,\%$ above those determined herein, across all supersaturations. This might be explained by our measurement site's larger distance to the Southern Ocean. As we will show below, air masses often traveled over Antarctica for extended times before reaching our measurement station, which might be connected to an increased wash out of CCN by precipitation along the way. The third column of Table 2 shows the ratio fraction of $N_{\mathrm{CN>90nm}}$ and $N_{\mathrm{CCN}}$ at different $SS$ to $N_{\mathrm{CN}}$ (based on the median values of the first column in Table 2). The values of $N_{\mathrm{CN>90nm}}/N_{\mathrm{CN}}$ and $N_{\mathrm{CCN,0.7\%}}/N_{\mathrm{CN}}$ are 0.06 and 0.64, respectively. This indicates, that the aerosol particles feature an Aitken mode dominance, as $94\,\%$ of the aerosol particles are smaller than

90 nm. Assuming a hygroscopicity parameter $\kappa$ of 0.8 for the coastal area of East Antarctica, taken from Pringle et al. (2010), the critical diameter $d_{crit}$ for $SS = 0.7\%$ was determined by means of the $\kappa$-Köhler theory to be $\approx 35$ nm. On the basis of this assumption 36 % the aerosol particles are smaller than roughly 35 nm. That is indicative for a high amount of freshly, secondarily newly formed aerosol particles, which form from precursor gases emitted from the ReactiveSouthern Ocean and

5 the Productive Zone as e.g., ammonia and DMS (see Section 2.3). The corresponding NPF events occurring during the passage of the air masses to the measurement site likely take place in the free troposphere (Fiebig et al., 2014; Quinn et al., 2017). Primary emitted natural aerosol particles that are known to occur in Antarctica from e.g., mineral dust (Wegner et al., 2015) or sea salt (Huang and Jaeglé, 2017; Yang et al., 2008; Wagenbach et al., 1998), are known to clearly exceed this size (Lamb and Verlinde, 2011). Unfortunately, we cannot examine the Aitken mode particles in much more detail, as our $PNSD$ data is in

the size range between 90 nm and 6.8 µm and hence only shows the accumulation and coarse mode particles. However, several other studies at coastal Antarctic sites report $PNSD$ measurements that show pronounced and dominant Aitken modes during austral summer (e.g., Asmi et al., 2010; O'Shea et al., 2017; Kim et al., 2017).

Panel D of Figure 6, 7 and 8 shows the regional analysis of the NAME footprints, as described in Section 2.3. It can give insights on the influence of the air mass origin on $N_{CN}$ and $N_{CCN}$. The regional analysis shows, that during the 10 days prior

to the measurements, air masses only have been influenced by the Antarctic continent, the Southern Ocean and the ReactiveProductive Zone region but not by South America or Africa. Thus, we can be confident, that we mainly measured pristine air masses and aerosol particles of a natural origin without much anthropogenic influence.

The contributions from Antarctica, the Southern Ocean and theReactiveProductive Zone region show a large variability. During 61 % of the measurement times, the air masses spent $\geq 90\%$ of the 10 days prior to their arrival at the measurement site

over the continental region. These times are called Continental Events (CEs) from now on. During CEs, we record only a low variability in the measured $N_{CN}$ and $N_{CCN}$. To illustrate this, the panels a) and b) of Figure 9 show a scatter plot and a box and whisker plot, respectively, displaying $N_{CN}$ verus the fraction of time that the respective air masses spent over the Antarctic region (continental fraction). All data from the three seasons are included. It can clearly be seen in both panels of Figure 9 that $N_{CN}$ scatters the least and reaches the lowest median value during CEs (note: only few data points exist for the low con-

tinental fractions, up to roughly 30%, making their median and percentiles statistically unreliable). During CEs, $N_{CN}$ rarely exceeds 475 cm$^{-3}$, with maximum values of 990 cm$^{-3}$, while 90 % of $N_{CN}$ (i.e., the 90 % percentile) cover a range from 170 to 475 cm$^{-3}$. The concentration ranges during CEs for $N_{CN}$, $N_{CN>90nm}$ and $N_{CCN}$ at all $SS$ are shown in the second column of Table 2. These concentration ranges can be assumed to be pristine Antarctic continental background concentrations during austral summer.

Vice versa, dDuring 39 % of the time the proportion of the ReactiveProductive Zone plus the Southern Ocean region was larger than 10 %, which we from now on call Marine Events (MEs). During MEs we record an enhanced variability in $N_{CN}$ and $N_{CCN}$. Also the precipitation, depicted in panel A of Figure 6, 7 and 8, shows a connection to MEs. Especially strong precipitation events only occur during certain most intense MEs affecting PE, e.g., on December 21 in 2013, January 18 in 2015 and January 30 in 2016, in line with the findings of Gorodetskaya et al. (2014) and Souverijns et al. (2018). These pre-

cipitation events significantly decrease lower $N_{CN}$ and $N_{CCN}$ on a time scale of some hours to one day, due to scavenging

and wet deposition. The minimum values that we report for $N_{CN}$ and $N_{CCN}$ were measured during these strong precipitation events. As the Antarctic region does not act as a significant source of water vapor (see katabatic meteorological regime in Section 2.1) it is self-explanatory that strong precipitation events only occur during MEs. But also the highest largest values for $N_{CN}$ and $N_{CCN}$ are only observed during MEs. The ReactiveProductive Zone and the Southern Ocean region potentially

represent source regions for primary and secondary formed aerosol particles. As already mentioned in Section 2.3 the region of the ReactiveProductive Zone can contribute to the Antarctic aerosol particle loading due to sea bird and penguin guano and microbiota occuring in open meltwater ponds and related to sea ice, all connected to and the release of ammonia that potentially contributes to the formation of new particles (Legrand et al., 1998)(Kyrö et al., 2013; Schmale et al., 2013; Dall'Osto et al., 2017) (Croft et al., 2016). Maybe more importantly, the Productive The Reactive Zone and the Southern Ocean region also emit

precursors for secondary aerosol particle formation as e.g., DMS, whose oxidation products sulfuric and methane sulfonic acid similarly contribute to NPF and have the ability to form aerosol particles that grow to CCN sizes (Liss and Lovelock, 2008). Also, these regions have the potential to contribute to the aerosol particle loading by primary emissions of sea salt particles due to blowing snow on sea ice surfaces (Huang and Jaeglé, 2017; Yang et al., 2008; Wagenbach et al., 1998) or bubble bursting from wave action (Lamb and Verlinde, 2011).

The time series of $N_{CN}$ in Figure 6, 7 and 8 often show a spontaneous increase during MEs of several thousand particles per $cm^3$. Figure 10 exemplarily shows such an event, that took place on December 6 in 2014. Between 7 and 10 a.m. $N_{CN}$ increased from $\approx 200$ to $\approx 6000\,\mathrm{cm}^{-3}$. This was accompanied by an increase of $N_{CCN}$. In total we detected 12 comparable events with an increase of $N_{CN}$ up to several thousand $cm^{-3}$ that all took place in a time frame between several hours and $\approx 1$ day. In the vast majority these events of increased $N_{CN}$ were followed by an increase of $N_{CCN}$ by a factor of roughly two, in

three cases even by an increase of more than a factor of 10. in a few cases even up to 15. This only holds for $N_{CCN}$ measured at SS between 0.2 %-0.7 %. $N_{CCN}$ measured at a SS of 0.1 % usually show a different trend and seem to be decoupled from the other measurements. Other studies at Antarctic sites report events of NPF during austral summer, e.g., Asmi et al. (2010) and Weller et al. (2015) at the Finnish research station Aboa and the German Neumayer station, respectively, which are both coastal sites. or Järvinen et al. (2013) even reported the observation of NPF at Dome C, a site in Central Antarctica. Median

growth rates of particles from NPF reported in these studies were $\approx 2.5\,\mathrm{nm/h}$ at Dome C throughout the year, and $3.4\,\mathrm{nm/h}$ and $0.6\,\mathrm{nm/h}$ for particles up to and above 25 nm, respectively, in the austral summer. At Aboa, variable growth rates were reported, ranging from $0.8\,\mathrm{nm/h}$ to $2.5\,\mathrm{nm/h}$ reported in Asmi et al. (2010) and from $1.8\,\mathrm{nm/h}$ to $8.8\,\mathrm{nm/h}$ derived in Kyrö et al. (2013), while growth rates were only and $\approx 1\,\mathrm{nm/h}$ for the two costal site of Neumayer (Weller et al., 2015). However, While it was also described that particles rarely grew to sizes larger than $\approx 25\,\mathrm{nm}$ at Neumayer (Weller et al., 2015), i.e., that

they do not reach sizes at which they can readily act as CCN, growth of newly formed particles into the CCN size range was reported for Aboa, likely due to precursor emissions from local meltwater ponds (Kyrö et al., 2013) or due to precursor gases advected to the site with marine/coastal air masses (Koponen et al., 2003). The surprisingly high growth rates observed at Dome C may be related to air masses that had picked up precursor gases for the formation of particulate matter over the Southern Ocean or the region defined as Productive Zone herein, and that were subsequently transported in the free troposphere followed

by descent over Antarctica (Fiebig et al., 2014). This likely is a process occurring widely spread in Antarctica, for which not

the availability of precursor gases but rather the photooxidative capacity regulates the connected NPF and particulate growth (Fiebig et al., 2014). Tropospheric NPF with subsequent growth therefore likely also explains the above described observations at the PE station.

Our measured $PNSD$s do not cover the size range of the nucleation and Aitken mode, however, in combination with measurements of $N_{CN}$ and $N_{CCN}$ we can state that comparably freshly formed particles originating from new particle formation events and subsequent growth were observed during our measurements. Our measurements at the PE station show, that these freshly formed aerosol particles seem to reach size ranges relevant for CCN activation and thus are climatically relevant.

## 3.2 Air mass origins and potential source regions

Figure 11 shows the spatial distribution of the PSCF calculated for $N_{CN}$, $N_{CN}$-$N_{CCN,0.7\%}$, $N_{CCN,0.7\%}$ and $N_{CCN,0.1\%}$. These four parameters represent concentrations of all particles (with a lower size cut at $3\,\mathrm{nm}$), particles in the size range up to $\approx 35\,\mathrm{nm}$, particles with sizes above $\approx 35\,\mathrm{nm}$ and the largest particles above $\approx 110\,\mathrm{nm}$, respectively. The analysis was done using the data of all three austral summer periods, which is a data set of approximately 230 days and a corresponding set of 88152 back trajectories. The $75\,\%$ percentile values of $N_{CN}$, $N_{CN}$-$N_{CCN,0.7\%}$, $N_{CCN,0.7\%}$ and $N_{CCN,0.1\%}$, on the basis of which the PSCF analysis was done, are 466, 184, 268 and 13, respectively. High values in the maps in Figure 11 indicate, which regions have a high potential to contribute to the $25\,\%$ of the highest number concentrations measured at the receptor site. The PSCF of $N_{CN}$ shows enhanced values over the region of the Southern Ocean, mostly between $60°\,\mathrm{S}$ and $40°\,\mathrm{S}$, but not over the Antarctic continental region. Hence, the marine region of the Southern Ocean is likely to be the dominant source region leading to an enhancement in $N_{CN}$ measured at PE, while the Antarctic continent itself is not likely to act as a particle source. This is in accordance with results discussed in Section 3.1, i.e., the low variability of measured number concentrations during CEs and the occurrence of high values of $N_{CN}$ observed for air masses connected to MEs.

$N_{CN}$-$N_{CCN,0.7\%}$ and $N_{CCN,0.7\%}$ are two complementary parameters, adding up to $N_{CN}$. The PSCF maps of $N_{CCN,0.7\%}$ and $N_{CN}$-$N_{CCN,0.7\%}$ show clearly distinct patterns, indicating that different source regions are likely to contribute to high concentrations of particles with sizes below and above $\approx 35\,\mathrm{nm}$. However, both share that their highest signals are again in the Southern Ocean between $60°\,\mathrm{S}$ and $40°\,\mathrm{S}$, however, at different longitudes. The PSCF of $N_{CN}$-$N_{CCN,0.7\%}$ (particles with sizes below $\approx 35\,\mathrm{nm}$) shows a large area of high signals between $40°\,\mathrm{W}$ and $60°\,\mathrm{E}$. When calculating transport times based on air mass back trajectories, an average transport time of 5.1 days from this area to PE station is obtained. The PSCF of $N_{CCN,0.7\%}$ (particles with sizes above $\approx 35\,\mathrm{nm}$) shows the largest area of high signals in a region between $140°\,\mathrm{W}$ and $80°\,\mathrm{W}$, for which the average transport time to the PE station is 8.8 days. These air masses usually travel either along the west wind drift through the Drake Passage and circumnavigate Antarctica before making landfall close to PE station, or they travel along the easterly winds over coastal East Antarctica till they reach the PE station. This is consistent with the predominance of the easterly wind component during synoptically driven MEs (Gorodetskaya et al., 2013; Souverijns et al., 2018). As already discussed in Section 3.1, the aerosol observed at the PE station features a dominant Aitken mode. This can be brought in line with the results disussed here. The aerosol particles that originate from the marine areas that show up dominantly in the PSCF

likely are mainly secondary aerosol particles that grow during the transport to the PE station. The size of the measured aerosol particles can be assumed to be a function of average transport time, corresponding to source regions for larger particles that are further away (when considering air mass traveling times).

The PSCF map for $N_{\text{CCN},0.1\%}$ differs from the others. Overall, values are lower, pointing towards a more uniformly distributed

origin of particles with sizes above $\approx 110\,\text{nm}$. But it should also be stressed that valules for $N_{\text{CCN},0.1\%}$ are generally low (see Table 2). The PSCF map shows almost no areas of enhanced values over the Southern Ocean, but several spots of comparably enhanced values show up along the coast of Antarctica, i.e., over the ReactiveProductive Zone region. The overlap congruence between these spots and the different shelf ice areas that are shown in Panel d of Figure 1 is noteworthy striking. The PSCF shows significantly increased values at the locations of the Ross, Ronne-Filchner and Amery shelf ice (1, 2 and 6 in Panel d

of Figure 1), and slightly increased values at the location of the Fimbul, West and Shakleton shelf ice (5, 7 and 8). Hence, the Antarctic shelf ice regions seem to be potential source regions for enhanced values of $N_{\text{CCN},0.1\%}$. We will elaborate on that further in the next section.

### 3.3 Hygroscopicity

For the data set presented here, the hygroscopicity parameter $\kappa$ can only be inferred for SS=0.1\%, for which the median

$d_{crit}$ was determined to be $110\,\text{nm}$. For higher SS, $N_{\text{CCN}}$ is above $N_{\text{CN}>90\text{nm}}$, i.e., $d_{crit}$ is below the lower size limit of the measured $PNSD$. Therefore, the hygroscopicity derived here is only valid for the low number of comparably large particles that are activated at 0.1\% (see Table 2). All $\kappa$ values from the three seasons have a median value of 1 and are shown in a histogram in Figure 12. These are generally high atmospheric $\kappa$ values covering a broad range between 0.5 and 1.6. Separate analysis of $\kappa$ for CEs and MEs results in $0.99 \pm 0.18$ (to which 64\% of all separate $\kappa$ values contribute) and $1.05 \pm 0.20$,

respectively. There is no clear difference in hygroscopicity of the here analyzed large particles of roughly $110\,\text{nm}$, independent of the time the air mass had been over the continent. This points towards common sources of these large particles for both, CEs and MEs, which are discussed in the following.

Large $\kappa$ values as those observed here typically are only found for particles consisting of inorganic substances (Petters and Kreidenweis, 2007). Particularly values of roughly 1 or above are only known to occur for sea salt. 0.95 was reported in

Wex et al. (2010) as the mean value for the sea spray signal in marine air masses, derived from a collection of ambient hygroscopic growth measurements. Zieger et al. (2017) give a value of 1.1 for inorganic sea salt particles at 90\% relative humidity, and Petters and Kreidenweis (2007) give mean values of 1.12 and 1.28 for NaCl, based on hygroscopic growth and CCN measurements, respectively. (It may be worth noting that $\kappa$ derived from hygroscopic growth typically is below that derived from CCN measurements, see Petters and Kreidenweis, 2007). The lower values we derived for $\kappa$ are too low to

originate from pure sea salt particles. In addition to But besides for inorganic compounds, marine aerosol particles may also contain internally mixed organic substances which reduce their hygroscopicity (Swietlicki et al., 2008). Secondarily formed aerosol particles of marine origin are a result of DMS oxidation and further reactions. They can be expected to contain sulfates, and Petters and Kreidenweis (2007) give a $\kappa$ value of 0.61 for ammonium sulfate, derived from CCN measurements. Overall, the range of $\kappa$ values we derived for particles with sizes of $\approx 110\,\text{nm}$ indicates that they are mostly composed of inorganic

substances. While the lowest $\kappa$ values we determined point towards a contribution of sulfate containing particles in the here examined particle size range of around 110 nm, the median $\kappa$ of 1 might even point towards a dominance of sea salt. This agrees with sea spray particles being generally larger in size, compared to particles formed during NPF and growth, so that they might contribute to particles in this size range. It also agrees with an observation made at the Aboa research station, where

sodium chloride was found for larger particles with sizes above 100 nm (Teinila et al., 2000).

Before we compare our results to literature, we want to mention that the uncertainty of the $\kappa$ values was inferred with a method based on Monte Carlos simulations as described in Herenz et al. (2017) and Kristensen et al. (2016). In this approach, uncertainties of input parameters needed for the calculation of $\kappa$ are combined, namely the uncertainties for particle sizing and counting as well as for the supersaturation adjusted in the CCNc. During Monte Carlo simulations, these parameters are

randomly varied within their uncertainty range during a large number of separate runs (10000 runs in this study) to yield the uncertainty of the derived $\kappa$ based on the uncertainty of the input parameters. This analysis shows that the uncertainties in our $\kappa$ values are in the same order than the variability of the values itself , i.e., the uncertainty in the derived $\kappa$ values can be explained based on measurement uncertainties. This allows no interpretation of the variability in $\kappa$ with respect to different air mass origins.

A few other studies already examined the hygroscopicity of Antarctic aerosol particles, as well as the impact of sea ice regions on it. During the PEGASO ship cruise that took place in the austral summer in 2015 in the proximity of the Antarctic Peninsula and the Filchner-Ronne ice-shelf, Dall'Osto et al. (2017) found an increased $N_{CN}$ (aerosol particles larger than 3 nm) within air masses with an origin over sea ice regions in comparison to air masses that originated over open water. Other studies further suggest, that sea ice regions efficiently emit sea salt aerosol particles, e.g., Huang and Jaeglé (2017); Yang et al. (2008);

Wagenbach et al. (1998). O'Shea et al. (2017) measured CCN at the Halley research station, $\approx 30$ km from the Weddell Sea on the Brunt Ice Shelf. They report a median $\kappa$ value of 0.66 during measurements in December for five different $SS$ (0.08, 0.2, 0.32, 0.41, 0.53 %). Also, they had an event of a median $\kappa$ value of 1.13 during two days when back trajectories indicate that air masses had passed over sea ice regions of the Weddell Sea. This is indicative for ice surfaces being able to emit aerosol particles with a high hygroscopicity and is in line with our findings. Pringle et al. (2010) applied the ECHAM-MESSy Atmospheric

Chemistry (EMAC) model to simulate the global distribution of $\kappa$ at the surface. That study results in values between 0.6 and 0.9 for Antarctic coastal areas and >0.9 for the Southern Ocean region. Asmi et al. (2010) measured the hygroscopicity of Antarctic aerosol particles at the Aboa station using a Hygroscopicity-Tandem Differential Mobility Analyser. They also found the Antarctic aerosol particles to be very hygroscopic with an average hygroscopic growth factor of 1.63, 1.67 and 1.75 for 25, 50 and 90 nm particles, respectively, at 90 % RH, which is larger than similar to the hygroscopic growth factor of ammonium

sulfate particles at 90 % RH (given as 1.64, 1.68 and 1.71 for these three different sizes in Asmi et al., 2010). Unlike these studies and our findings, Kim et al. (2017) report a lower particle hygroscopicity. Their results are based on CCN and $PNSD$ measurements that were conducted at the King Sejong Station in the Antarctic Peninsula between 2009 and 2015. For CCN measurements at a $SS$ of 0.4 % they found an annual mean $\kappa$ value of $0.15 \pm 0.05$, which, however, is the only time such low $\kappa$ values were reported for Antarctica.

Summarizing, we conclude that the few large aerosol particles we observe for sizes of and above $\approx 110$ nm may partially

originate from NPF and subsequent growth. In this respect, it should also be explicitly mentioned that cloud processing of particles also adds mass to those particles that are activated to cloud droplets (Ervens et al., 2018, and references therein), potentially aiding the growth of particles formed by NPF into here discussed size range. However, particulate mass added during cloud processing will not have $\kappa$ values above these of sulfates. Therefore, the majority of these aerosol particles in

the size range of $\approx 110\,\text{nm}$ likely consist of sea spray particles originating from the open ocean or even more so of sea salt particles emitted over sea ice regions, a statement we base on their comparably high $\kappa$ values. This fits to the results presented for $N_{\text{CCN},0.1\%}$ in Section 3.2, showing the marine areas in coastal proximity and especially the shelf ice regions as potential source regions.

## 4   Summary and conclusions

The data set presented here contains in situ ground-based aerosol particle data sampled at the Belgian Antarctic research station Princess Elisabeth (PE), in Dronning Maud Land in East Antarctica. During three austral summer seasons (2013-2016, each from December to February) we measured total aerosol particle number concentration and size distribution as well as the total CCN number concentration at 5 different supersaturations. An automatic weather station, located in the vicinity of the PE station, and a precipitation radar were used to gain further information about the meteorological conditions. The history

of the air masses arriving at the PE station was modeled by using the NAME dispersion model and the PSCF model based on HYSPLIT back trajectories.

$N_{\text{CN}}$ was found to range between 40 and $6700\,\text{cm}^{-3}$ with a median of $333\,\text{cm}^{-3}$. For particles being larger than $90\,\text{nm}$ ($N_{\text{CN}>90\text{nm}}$) we found a median concentration of $20\,\text{cm}^{-3}$. $N_{\text{CCN}}$ covers a range between less than $10\,\text{cm}^{-3}$ at SS=$0.1\%$ and $1300\,\text{cm}^{-3}$ for the highest largest $SS$ of $0.7\%$. The median values of $N_{\text{CCN}}$ for supersaturations of 0.1, 0.2, 0.3, 0.5 and

$0.7\%$ are 14, 81, 121, 177 and $212\,\text{cm}^{-3}$, respectively. All of the previous values are calculated on the basis of the entire measurement period of three austral summers. The ratios fractions of $N_{\text{CN}>90\text{nm}}/N_{\text{CN}}$ and $N_{\text{CCN},0.7\%}/N_{\text{CN}}$ indicate that $94\%$ and $36\%$ of the particles are smaller than $90\,\text{nm}$ and $\approx 35\,\text{nm}$, respectively. From this we conclude, that an Aitken mode dominated aerosol prevailed, that likely includes a significant amount of freshly, secondarily formed aerosol particles.

The fluctuations in $N_{\text{CN}}$ and $N_{\text{CCN}}$ can be associated with the history of the air masses and the precipitation measured at the

PE station. Both methods, the regional analysis on the basis of the NAME dispersion model as well as the PSCF analysis show, that high $N_{\text{CN}}$ values are directly linked to the advection of marine air masses, which we call marine events (MEs), having their origin in the region of the Southern Ocean. The occurrence of precipitation is also directly linked to the occurrence of MEs, as marine air masses are the only significant source of water vapor in Antarctica. Strong precipitation events caused the lowest $N_{\text{CN}}$ and $N_{\text{CCN}}$ values presented in this study, due to particle scavenging and wet deposition. Therefore, MEs showed

the lowest but also the highest particle concentrations measured. In contrast, when air masses had spent more than $90\%$ of the 10 days prior to arrival over the Antarctic continent, which are times we called continental events (CEs), measured $N_{\text{CN}}$ and $N_{\text{CCN}}$ values were comparably constant, and we assume these to be continental background concentrations during austral summer. The Antarctic continent itself was found to not act as a significant source of aerosol particles and CCN measured at

the PE station during these times. MEs and CEs occur 39 % and 61 % of the time, respectively.

The hygroscopicity of the CCNs could only be determined for measurements at SS=0.1 %, as the PNSDs could only be measured in a size range between 90 nm and 6.8 μm. The median $d_{crit}$ and $\kappa$ of the entire measurement period were determined to be 110 nm and 1, respectively. This high hygroscopicity, which is valid for the comparably small fraction of particles observed in the respective size range, which is in agreement with most of the other studies dealing with Antarctic hygroscopicity, and can mainly be attributed to the presence of mainly sea salt and likely, but to a minor fraction, sulfate aerosol particles. This is in agreement with the PSCF analysis, for which the Antarctic ice shelf areas were found to cause elevated values for particles with sizes above ≈ 110 nm, again pointing to sea salt aerosol particles. These particles could have been released and formed from snow particles from surface snow layers by sublimation, e.g., during periods of high wind speed when fresh snow is available and winds are high enough to cause drifting or blowing snow (Gossart et al., 2017) or else may originate from sea spray directly.

Although this is to our knowledge the most comprehensive set of CCN data in the region of East Antarctica, it is limited on the austral summer seasons. To get a complete picture of CCN properties in East Antarctica, full time measurements throughout the whole year are needed, together with PNSD measurements covering diameters down to a few some nanometer. This would enable more in-depth investigations of new particle formation and particle hygroscopicity in different size ranges. However, the data presented here increases our knowledge of aerosol particle and in particular CCN properties in Antarctica.

*Acknowledgements.* This work was funded by the EU FP7-ENV-2013 program "Impact of Biogenic vs. Anthropogenic emissions on Clouds and Climate: towards a Holistic UnderStanding" (BACCHUS), project number 603445 and by the Belgian Science Policy Office under contract EA/34/1B and under grant number BR/143/A2/AEROCLOUD. We thank Wim Boot, Carleen Reijmer and Michiel Van den Broeke (Utrecht University, Institute for Marine and Atmospheric Research, The Netherlands), Alexandra Gossart and Nicole van Lipzig (KU Leuven) for the data of the Automatic Weather Station. We also thank Niels Souverijns and Nicole van Lipzig (KU Leuven) for the provision of the precipitation data. The authors acknowledge the NOAA Air Resources Laboratory (ARL) for the provision of the HYSPLIT transport and dispersion model and would like to thank the UK Met Office for the use of the NAME dispersion model and the STFC JASMIN computer for hosting the model.

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

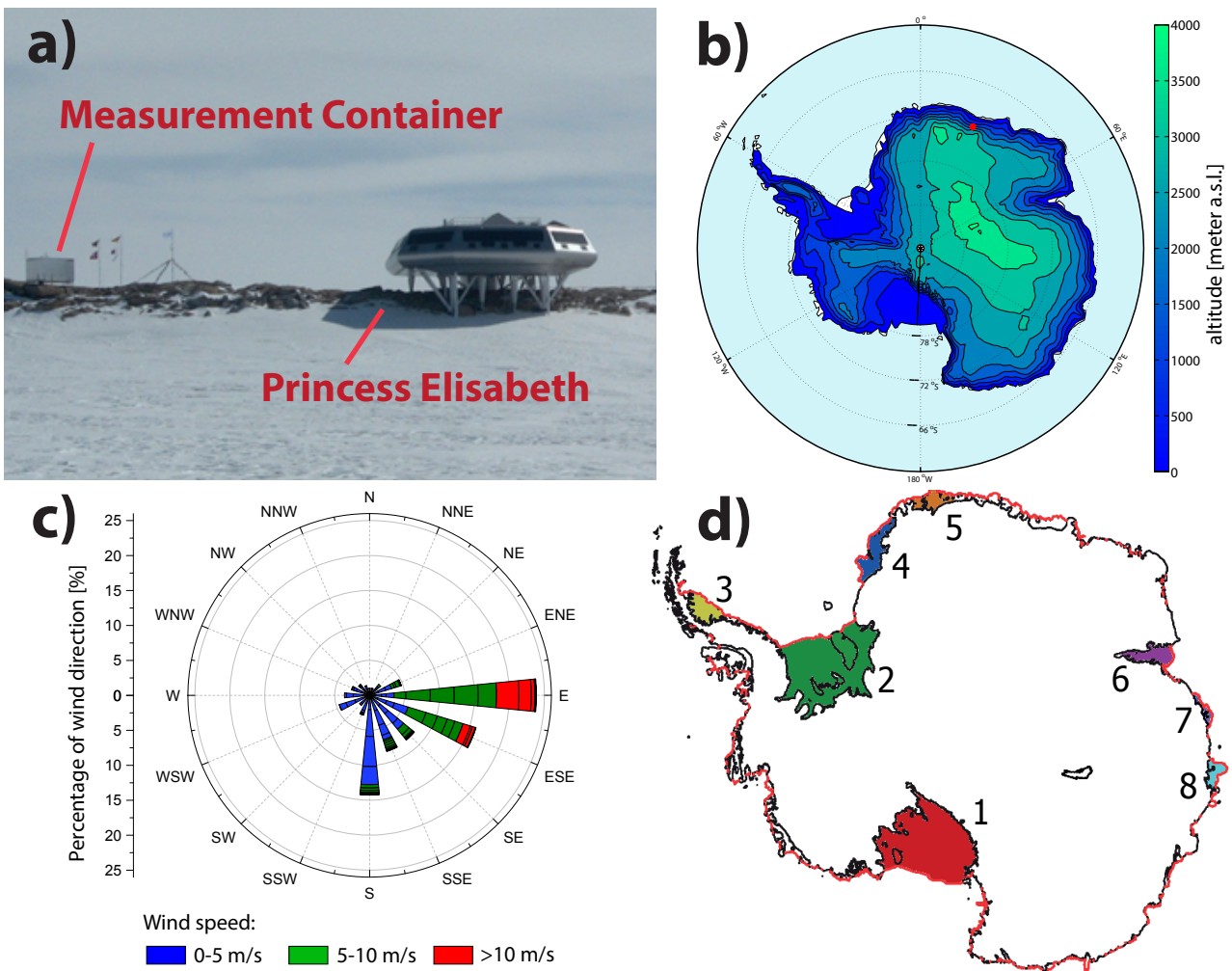

**Figure 1.** a) Picture of the Princess Elisabeth research station and the measurement container, where the aerosol measurements were performed, both located on the Utsteinen Nunatak ridge (view from east-south-east ESE). b) Topographic map of Antarctica, the red dot shows the location of PE. This map was done using the Matlab mapping package M_Map. c) Wind direction and wind speed depicted as a wind rose for the third measurement period (December 18, 2015 to February 20, 2016). d) Map showing the location of the largest shelf ice regions in Antarctica: 1) Ross 2) Ronne-Filchner 3) Larsen C 4) Riiser-Larsen 5) Fimbul 6) Amery 7) West and 8) Shackleton shelf ice. The black line represents the coast line and the red line represents the ice edge. This map was created using Matlab and Antarctic Mapping Tools (Schaffer et al., 2016; Greene et al., 2017).

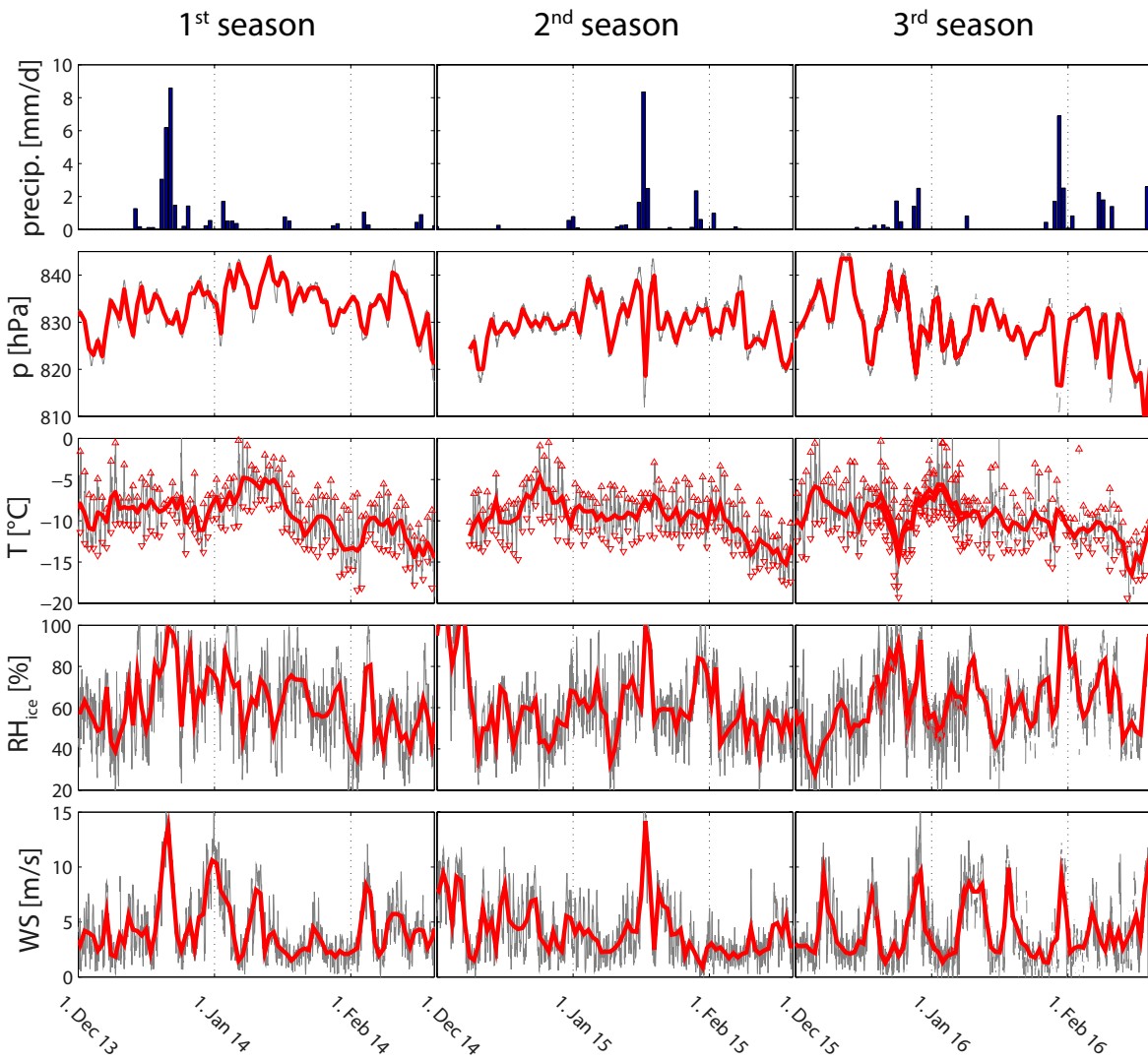

**Figure 2.** Time series of hourly (gray) and daily (red) mean values for temperature, pressure, relative humidity with respect to ice (RH) and wind speed (WS) measured by the AWS. Maximum and minimum temperature values are shown as triangles. The daily precipitation measured by the precipitation radar is shown as bars.

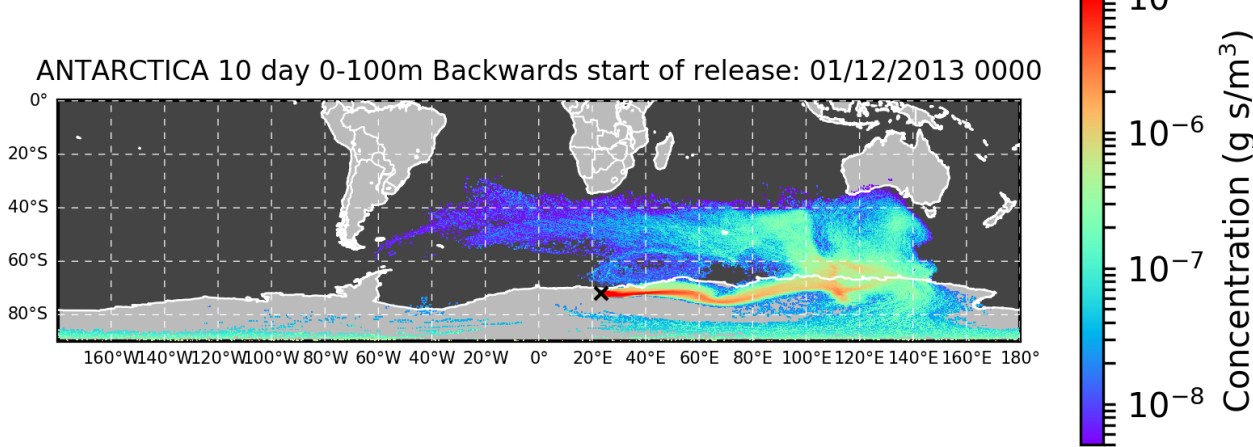

**Figure 3.** NAME dispersion model 10 day backwards footprint.

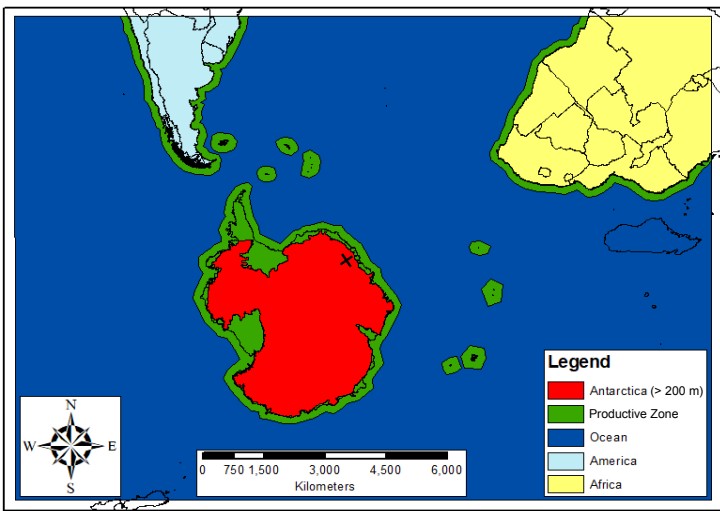

**Figure 4.** 5 different regions (Southern Ocean, Antarctic inland plateau, South America, Africa and ReactiveProductive Zone) that are used to track the percentage residence time in each region before arriving at PE from the NAME footprints.

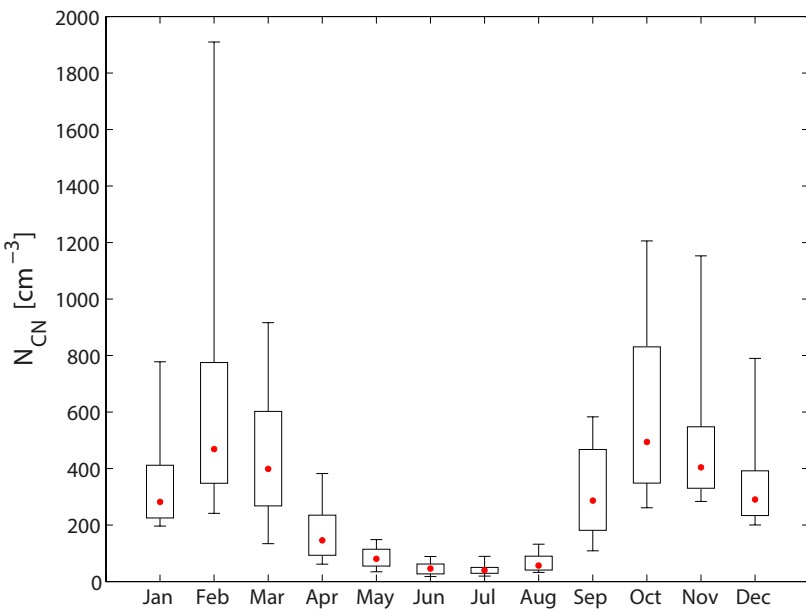

**Figure 5.** Box plot of monthly median values (red dots), interquartile range (black box) and $10\%$ and $90\%$ percentile (black bars) of $N_{CN}$ measured at the PE station between 2012 and 2016.

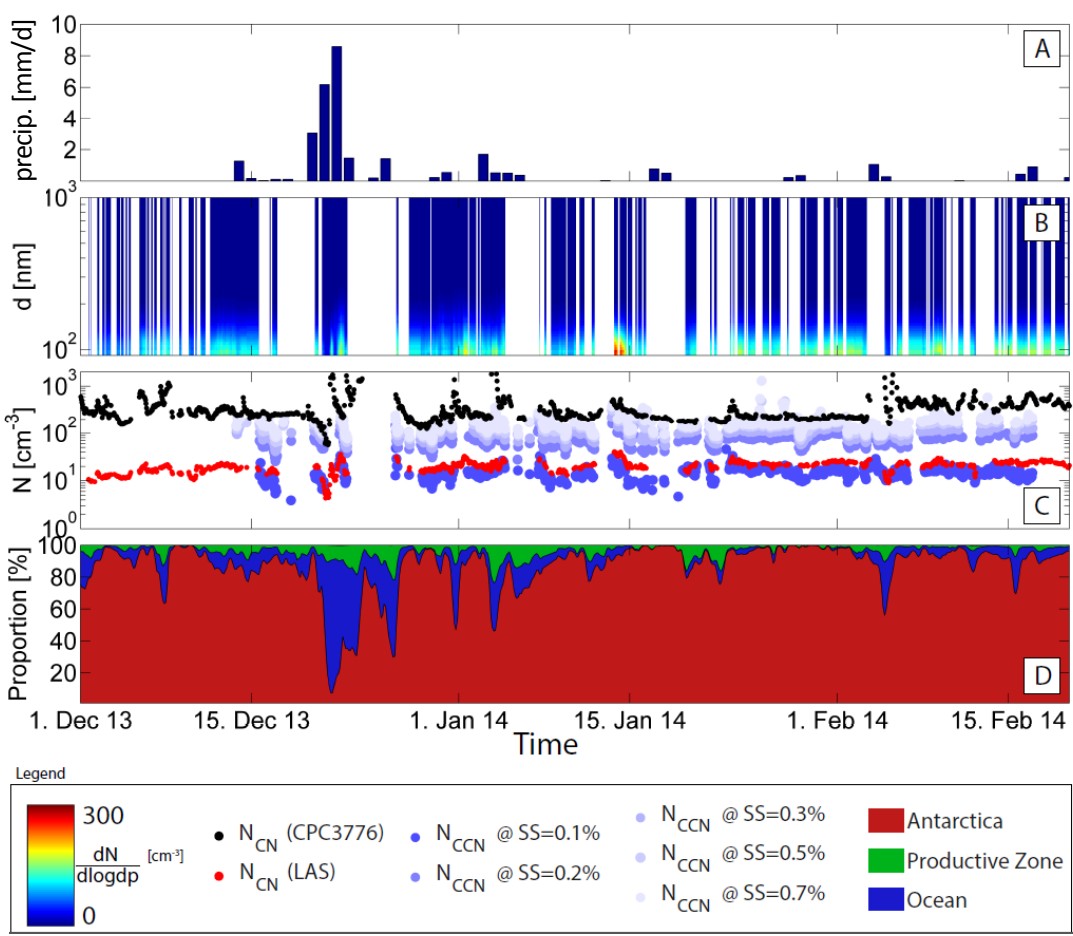

**Figure 6.** Time series of the first season (December 2013 to February 2014) of A) daily precipitation shown as bars (same data as in Figure 2), B) $PNSD$, depicted between 90 nm and 1 µm, C) $N_{CN}$ measured by the CPC in black, $N_{CN}$ measured by the LAS (integrated concentration between 90 nm and 6.8 µm) in red and $N_{CCN}$ measured by the CCNc at $SS$ between 0.1 and 0.7 % in different blue colors, D) proportion of residence of the air masses over the Antarctic continent (red area), the ReactiveProductive Zone (green area) and the Southern ocean (blue area) areas during the past ten days (based on the NAME model footprints).

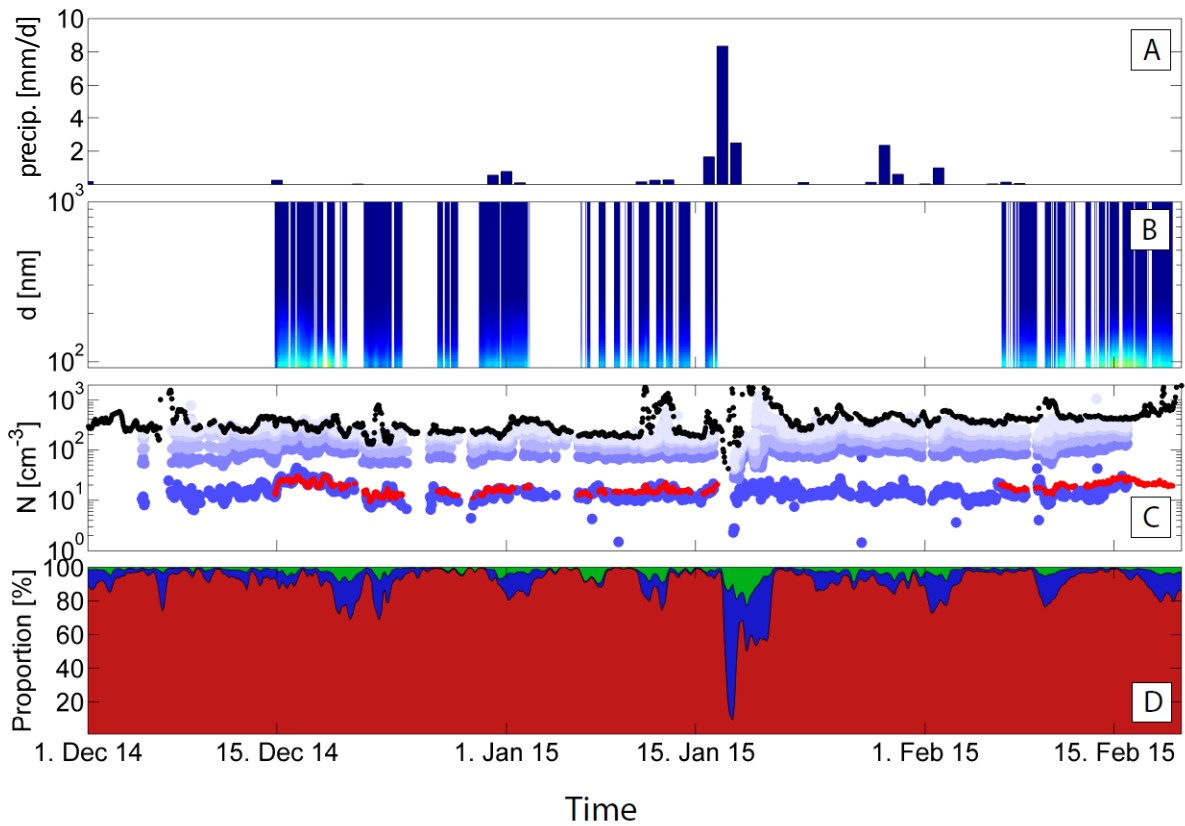

**Figure 7.** Time series of the second season (December 2014 to February 2015). For further details see caption of Figure 6.

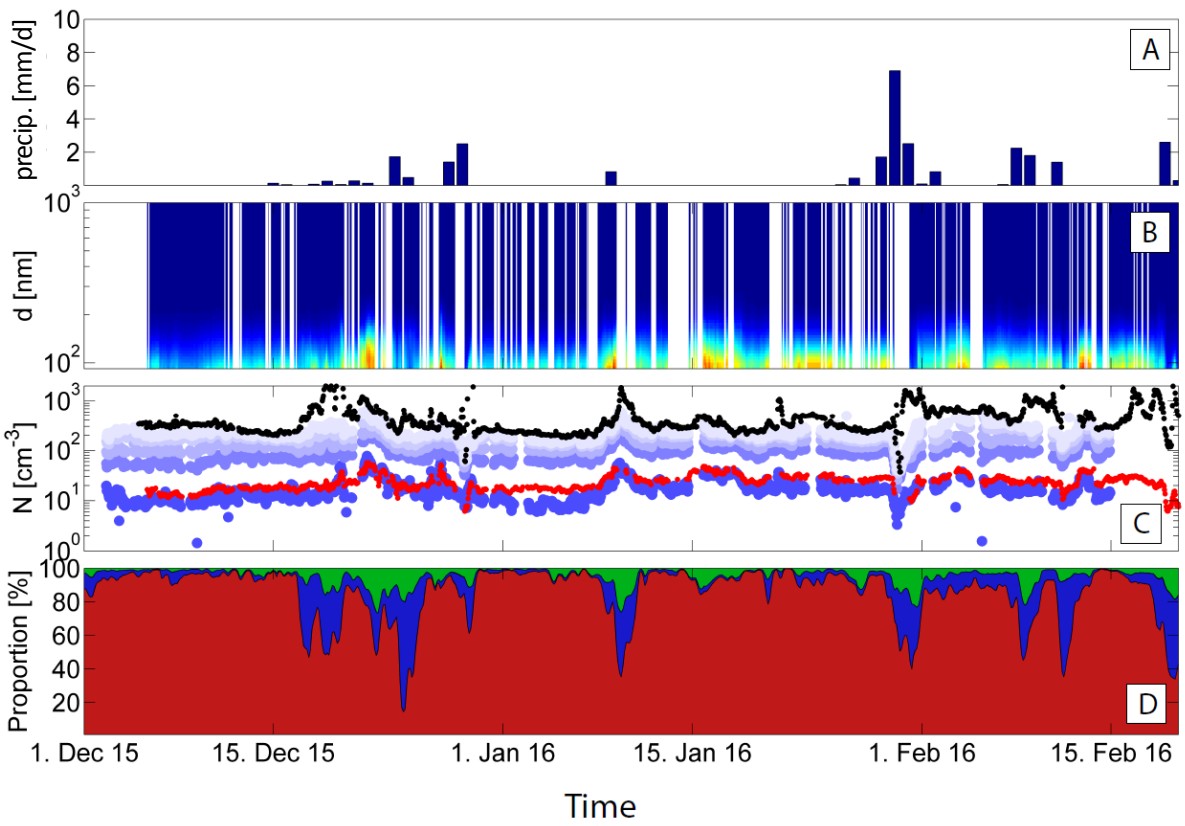

**Figure 8.** Time series of the third season (December 2015 to February 2016). For further details see caption of Figure 6.

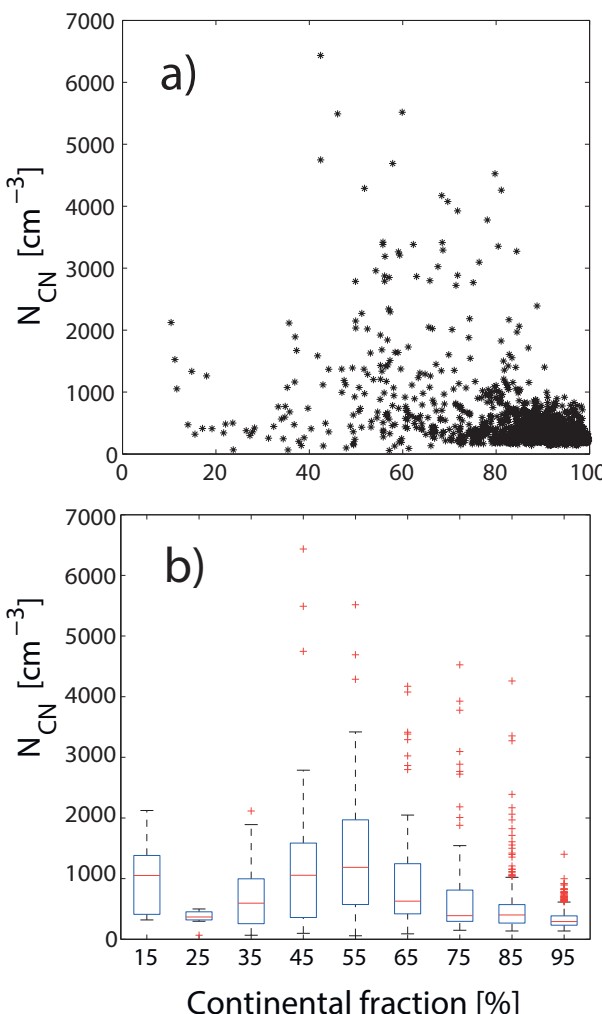

**Figure 9.** Connection between $N_{CN}$ and the occurrence of continental air masses. While panel a) shows the data separately, panel b) gives a box and whisker plot with median values and the interquartile range (blue box).

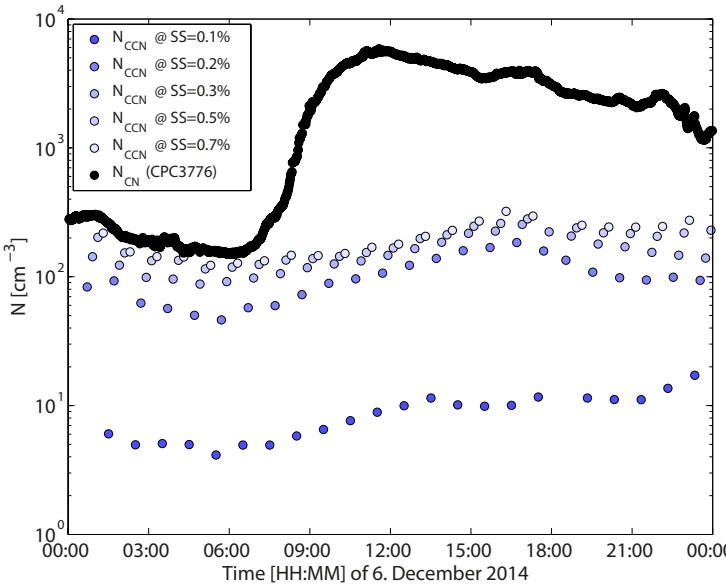

**Figure 10.** $N_{\mathrm{CN}}$ and $N_{\mathrm{CCN}}$ during an event of new particle formation at the PE station.

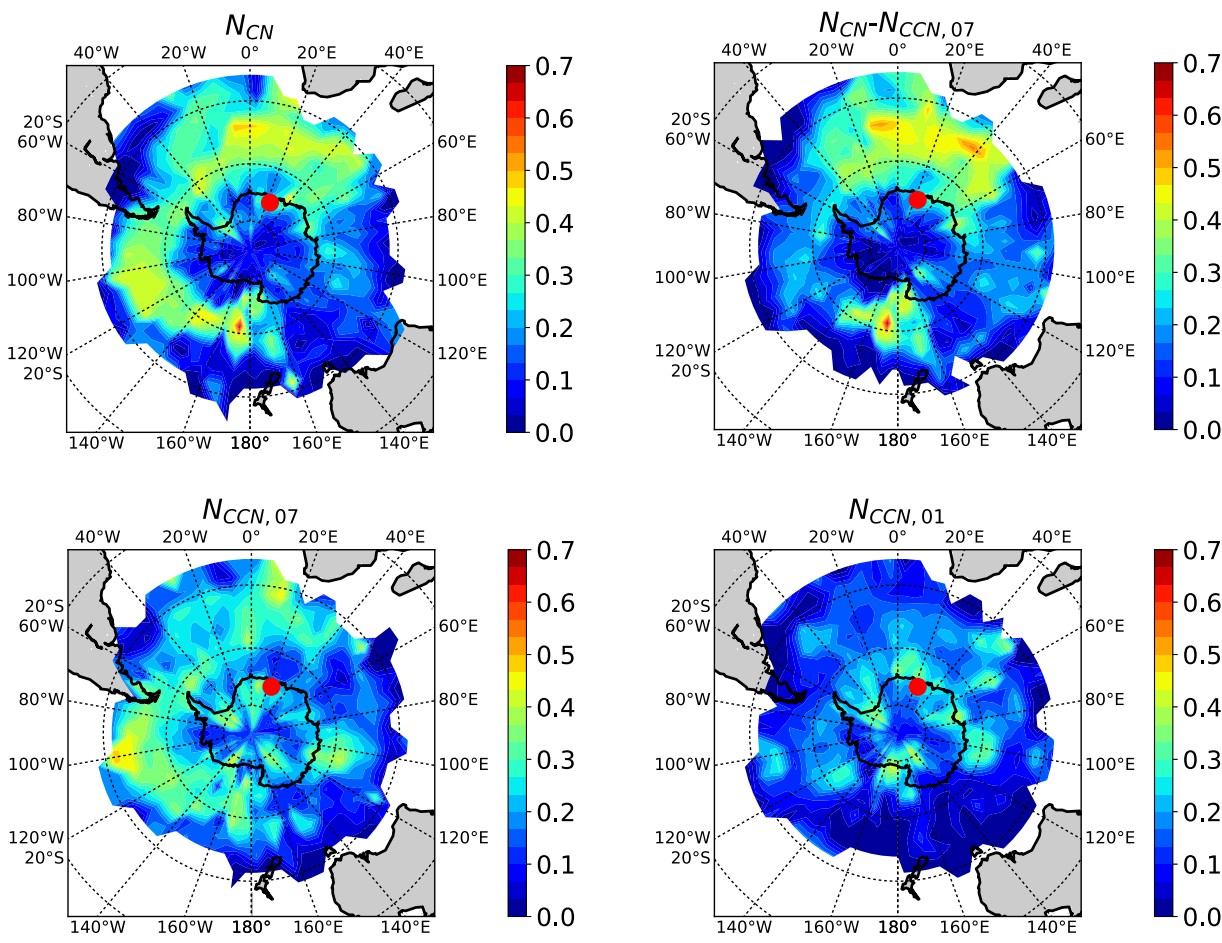

**Figure 11.** PSCF results that are plotted over a map of Antarctica for $N_{CN}$, $N_{CN}$-$N_{CCN,0.7\%}$, $N_{CCN,0.7\%}$ and $N_{CCN,0.1\%}$. The colorbar indicates the value of the PSCF.

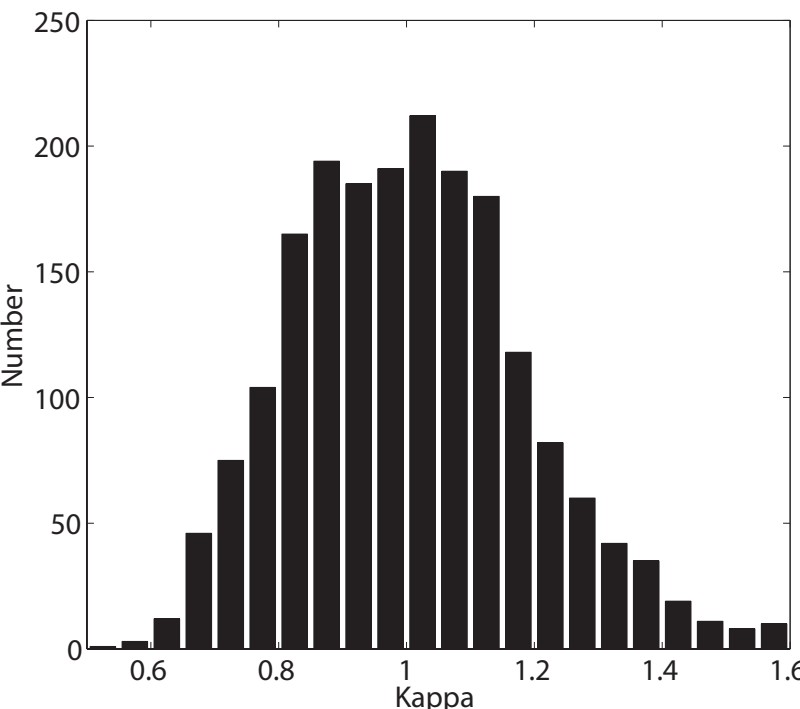

**Figure 12.** Histogram showing the 2171 $\kappa$ values of all three seasons, derived for a supersaturation of 0.1 %, for which the median $d_{crit}$ was determined to be 110 nm.

**Table 1.** Basic meteorological parameters (temperature, pressure, relative humidity with respect to ice, wind speed and precipitation) measured by the AWS and the precipitation radar. Shown are the mean, standard deviation, minimum and maximum values based on hourly mean values, in case of precipitation daily mean values, for the three measurement periods (each from 1 December to 20 February).

| | 2013/2014 | 2014/2015 | 2015/2016 |
|---|---|---|---|
| variable | Mean (std), Min, Max | Mean (std), Min, Max | Mean (std), Min, Max |
| Air temperature [°C] | -9.4 (3.1), -19.3, 0.5 | -9.7 (2.7), -17.6, -0.5 | -10.1 (2.7), -20.5, 2.3 |
| Pressure [hPa] | 833.4 (5), 817.3, 844.3 | 829.7 (4.7), 812, 843.5 | 828.6 (6.6), 807, 845.3 |
| RH [%] | 61.9 (18.6), 13.1, 100 | 58.9 (17.2), 14, 100 | 64.3 (18.2), 14.7, 100 |
| Wind speed [m/s] | 4.35 (2.87), 0.13, 16.21 | 4.26 (2.89), 0.03, 22.59 | 4.21 (2.99), 0, 16.6 |
| Precipitation [mm/d] | 0.38(-), -, 8.6 | 0.24 (-), -, 8.3 | 0.35 (-), -, 6.9 |

**Table 2.** Overview showing $N_{CN}$, $N_{CN>90nm}$ and $N_{CCN}$ at different supersaturations, given as median (and 10 % and 90 % percentiles in brackets) in column 1 for all data, in column 2 for CEs (continental events, based on the regional analysis of the NAME model output). Column 3 shows the ratio fraction of $N_{CN>90nm}$ and $N_{CCN}$ to $N_{CN}$ (based on the median values of column 1).

| Parameter | Median concentration (10 %, 90 % percentile) [$cm^{-3}$] | Median concentration during CEs (10 %, 90 % percentile) [$cm^{-3}$] | $N_{CN}$ (LAS)/$N_{CN}$ (CPC) or $N_{CCN}$/$N_{CN}$ (CPC) |
|---|---|---|---|
| $N_{CN}$ (CPC) | 333 (206, 893) | 292 (205, 474) | - |
| $N_{CN>90nm}$ (LAS) | 20 (14, 29) | 20 (14, 29) | 0.06 |
| $N_{CCN,0.1\%}$ | 14 (10, 23) | 14 (10, 21) | 0.04 |
| $N_{CCN,0.2\%}$ | 81 (56, 110) | 79 (58, 105) | 0.24 |
| $N_{CCN,0.3\%}$ | 121 (90, 168) | 120 (95, 161) | 0.36 |
| $N_{CCN,0.5\%}$ | 177 (125, 260) | 177 (133, 232) | 0.53 |
| $N_{CCN,0.7\%}$ | 212 (138, 326) | 210 (150, 292) | 0.64 |