# Peer review of "CCN measurements at the Princess Elisabeth Antarctica Research Station during three austral summers"

_Atmospheric Chemistry and Physics, 2018_

## Referee Comment (RC1) · Anonymous Referee #3 · 13 May 2018

**Review of manuscript ACP-2018-245**

Presented in this manuscript are the data from the measurements of total concentration and size distribution of aerosols and CCN concentrations at several different supersaturations in three austral summer seasons at an Antarctic site. Despite their importance, such measurements have been relatively rare due to remoteness and harsh environment of the Antarctic region. In that sense, this manuscript can be a valuable addition to Antarctic aerosol dataset. This manuscript is well organized and easy to follow. I think that this manuscript can be accepted for publication with some minor revision that I indicate below.

**Major comments:**

As the authors did, the critical diameter for a given supersaturation can be estimated by simply comparing CCN concentration at the given supersaturation to particle size distribution and finding the diameter at which the integrated particle concentration matches with the CCN concentration. From the critical diameter for the given supersaturation, kappa can also be estimated as the authors did. In this approach, however, the important assumption is that all particles were of the same chemical composition, i.e., internally mixed. The authors should explain how well this assumption can be justified for the Antarctic aerosols. Furthermore, particle size distribution measurement should be made for dried particles to be able to legitimately designate the matching diameter as the critical diameter. However, the authors seem to have measured ambient particle size distribution, not dried particle size distribution. The RH in this region was occasionally very high (e.g., 1 Dec. 2014). The RH of the sample air would have been lower than the ambient RH because the temperature of the container was higher but I doubt that particles were completely dried in the sample tubing. Therefore, the authors could have overestimated the critical diameter and underestimated kappa. The authors should make some detailed discussion on this issue. To note is that my comments is to the version of the manuscript posted for ACPD.

The authors showed representative percentile concentrations only for CEs in Table 2. Adding those for MEs (perhaps excluding precipitation days) would be informative, giving some quantitative measure of the difference between CEs and MEs.

The authors did not analyze the variability of kappa with respect to different air mass origins due to high uncertainty. However, the authors can present some representative kappa values for CEs only and MEs only. They may not show a meaningful difference but such result would support the argument that large particle concentration did not have significant dependence on air mass origin.

P5, L16: It was mentioned that necessary corrections were very high in the two LAS channels below 100 nm when the authors used ammonium sulfate particles. What about 100 nm particles? I also wonder if the differences between LAS and SMPS were similar for ammonium sulfate particles and PSL particles of the same selected size of 100 nm.

The estimated kappa values were compared with comparable measurements in some other studies. The same should be done for total particle concentrations and CCN concentrations to illustrate how these concentrations fair with other measurements in the Antarctic region.

**Minor comments:**

P2, L24: positive → increasing

P5, L29: also → at; Apparently the authors observed CCN at 1% SS. How often? What does "adjusted" mean here?

P6, L1-L2: It does not read smoothly. Rewrite this part.

P7, L27: remove one 'influence.'

P8, L8: 88152 is not a multiple of 15. Why?

P8, L9: here and at other places, the words "trajectory segment endpoints" could be misleading. In fact, the endpoint is the starting point of the air mass movement. Hopefully the authors can suggest better words.

P9, L20: panel C Figure 6, 7 → panel C of Figures 6, 7

P12, L34: a κ value of e.g., κ = 0.61 → a κ value of 0.61

P13, L5: I am not sure what you mean by "in the same order than the variability of the value itself." Rewrite it.

P13, L14: during which → when

P13, L21: "ammonium sulfate" can be replaced by "sodium chloride" if kappa is 1.75.

P14, L34: the here presented data → the data presented here

Figure 4: Explain the unit "g s/m3."

Figure 6: The legend of color scale in the bottom panel should be dN/dlogD instead of dN.

---

## Referee Comment (RC2) · Anonymous Referee #1 · 6 Jun 2018

Review of "CCN measurements at the Princess Elisabeth Antarctica Research Station during three austral summers" by Paul Herenz et al.

This study describes the aerosol particle number concentration (CN) and cloud condensation nuclei number concentrations (N_CCN) at various supersaturations at the East Antarctic research station Princess Elisabeth. Also, limited particle number size distribution data, aethalometer and meteorological data are available. The authors focus on the austral summer measurements between 2013 and 2016. They analyze CN and CCN data additionally with two methods: (a) a particle dispersion model to quantify the fractional contribution of air masses from certain source regions that the authors

define, and (b) with a Hysplit based potential source contribution function model to illustrate the potential source regions for the highest 25 % occurrences of CN and N_CCN. The study finds that N_CCN can vary by 2 orders of magnitude reaching up to 1300 cm-3 for supersaturations (SS) between 0.1 and 0.7%. The Aitken mode is the dominant particle mode and 94 % of the particles are < 90 nm. Most air masses reside relatively long over the ice sheet before reaching the station, while a minor fraction arrives more directly from the Southern Ocean. The latter types of air masses bring higher CN and N_CCN concentrations. The contributions to the hygroscopicity parameter kappa, which ranges widely, are discussed, but no conclusion is derived.

There is no doubt that this data set needs to be published, because it furthers our understanding of CN and N_CCN in eastern Antarctica, a region that is heavily undersampled. The manuscript is clearly structured and generally well written. Before the manuscript can be published, however, it will need to undergo major revisions which are pointed out in the following with subsequent more specific comments. Technical and some specific edits can be found in the attachment.

First, it is unclear why the authors use the NAME dispersion model to create the footprints with the regional classification and the PSCF based on Hysplit for the potential source contributions. These are two different models used for similar purposes. There is no discussion in how far the models produce similar or deviating results, especially in light of the presumably different meteorology (GDAS vs UM meteorological field data). Either, option a), there needs to be a very clear explanation why two different models are used, including a comparison between them to show that the results are comparable, or, option b), either the footprint analysis needs to be done with Hysplit or the PSCF needs to be done with NAME. This might require slight modifications of how the models are run.

Second, the regional classification does not make sense and is misleading. The authors imply that the "Reactive Zone" is the largest source of particles and that ammonium and organic nitrogen might be an important contribution to particle mass. Both

aspects are likely wrong:

a) The larger source of particles will be the open ocean through sea spray production (Quinn et al., 2017), because this area is much larger than the "Reactive Zone". Generally the particles are acidic and not neutralized from ammonia emissions. Those have a limited regional effect.

b) The animal colonies are an important source of ammonia and lead to secondary particle mass enhancement. But (a) the source is not that important (Riddick et al., 2016; Riddick et al., 2012) compared to the Southern Ocean sea spray emissions, and (b) the authors do not show whether the ammonia emissions are responsible for the increase in CN and N_CCN at PE because they don't have chemical composition data and the kappa value discussion is too vague. Also, ammonia emissions do not necessarily lead to an increase in particle number concentration. They typically lead to the increase of particle mass. This has been shown by e.g., Schmale et al. (2013). By the way, it is unclear why the authors use a references from an Arctic study (Croft et al. 2016) when there is a study on ammonia emissions from sea birds and seals in the Southern Ocean (Schmale et al., 2013).

c) Including the permanently and seasonally covered sea ice areas into the "Reactive Zone" for the reasons given, i.e. organic nitrogen and secondary particle formation, is highly misleading. The contribution of organic nitrogen as found by Dall'Osto et al. 2017 is miniscule. The finding is interesting but the relevance for N_CCN is far from being understood and not large. Also, the authors consider the sea ice zone as static, without taking into account the year to year variability. A proper analysis of the influence of the sea ice region requires using satellite images of the sea ice and marginal ice zone borders for each of the seasons considered in this work. Again, conclusions can be misleading.

d) The coastal areas around the continents and islands are another component of the "Reactive Zone". The reason the authors give is the enhanced chlorophyll concentra-

tions. Here again, it does not make sense to statically classify an area as chlorophyll-rich throughout the considered seasons. In addition to the fact that blooms will not be active to the same degree throughout the whole summer season and every season and year, there are plenty of other regions in the Southern Ocean where enhanced chlorophyll is typically observed (Valente et al., 2016). Those regions should be included as well. Again, satellite images of chlorophyll-a concentrations for the respective periods would have to be used to come to a trustworthy conclusion.

e) The term "Reactive Zone" is uninformative. The authors mainly refer to regions with enhanced microbial productivity, hence calling it "productive zones" is more appropriate. Furthermore, in the entire manuscript phytoplankton is not mentioned once. The reason for chl-a detection and VOC emissions that can be converted into particles (e.g. DMS to MSA) is the presence of phytoplankton. This link needs to be included explicitly in the manuscript to describe particle sources in the Southern Ocean and Antarctica.

Specific comments:

PSCF: In how far is the altitude of air trajectories included in the PSCF? An air mass that travels aloft, will likely not be influenced by surface emissions, hence the PSCF result can be misleading if any trajectory altitude is considered. This requires clarification and potentially recalculation. Also, on p. 8, l. 24 the authors state that the criteria are empirical without clarifying where the empirical evidences comes from.

NAME and PSCF: It seems that mostly surface sources of CN and N_CCN have been considered. Over the Southern Ocean and Antarctica it is assumed that a significant fraction of particles comes from particle formation in the free troposphere e.g., in the outflow of clouds (Quinn et al., 2017 and references therein). This is not considered at all, but needs to be included into a comprehensive discussion of particle sources.

Growth Rate (GR) discussion: On p. 11, the growth rate discussion seems incomplete. There is no connection to the observation discussed in the published literature (Aboa, Dome C) and the observations at PE (no GR was determined at PE). Furthermore,

the reader is left puzzled by the statement that the GR was higher at Dome C than at coastal sites. An explanation or at least discussion of this counter-intuitive observation needs to be included. Kappa: There is no explicit discussion of the sources of uncertainty for the derivation of the hygroscopicity parameter kappa (p. 12, l. 32 f), just the reference to the Monte Carlo method. More details are necessary.

Origin of particles > 110 nm: The authors list 2 options (p. 13, l. 21ff): NPF and subsequent growth, and sea spray / sea salt particles emitted over the sea ice region. Based on the previously cited literature on GR (1 – 2.5 nm/h) a sustained growth over a period of 40 to > 100 hours would be needed which is unlikely given the lack of condensable gases. Why is there no discussion about cloud-processed particles and emissions from the open ocean? Those are two mechanisms likely to produce accumulation mode particles. Those mechanisms should also be considered in the discussion on p. 12, l. 1-2.

Size of particles as function of transport time: This argument brought forward on p. 12, l. 2ff is not convincing keeping in mind the comment above. Particle size can be a function of the emission source and the cloud-processing as well. I recommend plotting the particle size vs transport time to back up this statement.

Figure 3: The purpose of the figure is unclear. It does not contribute any relevant information.

Summarizing, we conclude that the few large aerosol particles we observe for sizes of and above $\approx 110$ nm may partially originate from NPF and subsequent growth. However the majority of these aerosol particles likely consist of sea spray particles or even more so of sea salt particles emitted over sea ice regions. This fits to the results presented for $N_{CCN,0.1\%}$ in Section 3.2, showing the marine areas in coastal proximity and especially the shelf ice regions as potential source regions.

**4 Summary and conclusions**

[revised manuscript text omitted]

computer for hosting the model. We thank Wim Boot, Carleen Reijmer and Michiel Van den Broeke (Utrecht University, Institute for Marine and Atmospheric Research, The Netherlands) for the data of the Automatic Weather Station. We also thank Irina Gorodetskaya and Niels Souverijns (Catholic University of Leuven, Belgium) for the provision of the precipitation data.

[revised manuscript text omitted]

---

## Author Comment (AC1) · 24 Aug 2018

**Review of manuscript ACP-2018-245**

We thank reviewer 1 for reviewing our manuscript and for giving comments and suggestions. Our answers are given in blue, below, while the original text of the review was kept in black. (Page and line numbers refer to the new version of the manuscript which was attached at the end of this answer, in which deleted text is given in yellow while newly added text is given in blue).

Presented in this manuscript are the data from the measurements of total concentration and size distribution of aerosols and CCN concentrations at several different supersaturations in three austral summer seasons at an Antarctic site. Despite their importance, such measurements have been relatively rare due to remoteness and harsh environment of the Antarctic region. In that sense, this manuscript can be a valuable addition to Antarctic aerosol dataset. This manuscript is well organized and easy to follow. I think that this manuscript can be accepted for publication with some minor revision that I indicate below.

**Major comments:**

As the authors did, the critical diameter for a given supersaturation can be estimated by simply comparing CCN concentration at the given supersaturation to particle size distribution and finding the diameter at which the integrated particle concentration matches with the CCN concentration. From the critical diameter for the given supersaturation, kappa can also be estimated as the authors did. In this approach, however, the important assumption is that all particles were of the same chemical composition, i.e., internally mixed. The authors should explain how well this assumption can be justified for the Antarctic aerosols.

Thank you for this remark. You are correct: the approach we use assumes that all similarly sized particles are of the same chemical composition, i.e., internally mixed. But we can determine  $\kappa$  only for comparably large particles of roughly 110 nm. Total particle number concentrations of particles from that size on are low (we found a median value of 14 cm-3 - see our Table 2). And we do find a broad distribution of  $\kappa$  values (see our Figure 12). Going into any more detail about possible external mixtures on the basis of our data will not be possible. But we now state clearly in the text, that we make the assumption you mention here:

page 8, line 8: "This approach, however, does assume that all particles of roughly the size of  $d_{crit}$  have the same chemical composition, i.e., are internally mixed. Therefore the derived  $\kappa$  values will only give a rough information on the chemical composition of the examined aerosol, which, however, still can be useful in interpreting the origin of the observed aerosol particles."

Furthermore, particle size distribution measurement should be made for dried particles to be able to legitimately designate the matching diameter as the critical diameter. However, the authors seem to have measured ambient particle size distribution, not dried particle size distribution. The RH in this region was occasionally very high (e.g., 1 Dec. 2014). The RH of the sample air would have been lower than the ambient RH because the temperature of the container was higher but I doubt that particles were completely dried in the sample tubing. Therefore, the authors could have overestimated the critical diameter and underestimated kappa. The authors should make some detailed discussion on this issue. To note is that my comments is to the version of the manuscript posted for ACPD.

The temperature of both the container and the LAS was monitored and was between 15°C and 25°C. The temperatures outside generally were below -5°C, and typically lower when RH was high (-7°C or below). A dew-point of -7°C and container temperatures of 15°C result in a RH of 21%. In parallel to the instruments used in this study, also a nephelometer was used, which was also similarly fed with air from the outside and which measured the relative humidity in the measurement chamber. Values it reported showed a maximum RH of 20%, i.e., close to the value derived above, and mostly the RH in the nephelometer was even below 10%. This, together with past experience (both in the field as well as laboratory work on hygroscopicity done by some of the authors) makes us confident that the aerosol measurements we did were done for sufficiently dry particles. Moreover, in the CCNc the aerosol is wetted anyway, so in this case drying is not needed. We added the following information on this to the text:

page 6, line 13: "While no aerosol drying was installed in front of the LAS, ambient humidities and temperatures together with temperatures inside the container and the LAS were such that it can safely be assumed that relative humidities in the aerosol sampled for size distribution measurements were below 20%. "

The authors showed representative percentile concentrations only for CEs in Table 2. Adding those for MEs (perhaps excluding precipitation days) would be informative, giving some quantitative measure of the difference between CEs and MEs.

We added a second part to Fig. 9, which is the figure in which the total particle number concentrations are shown as a function of the continental fraction. This second part shows the same data but as box and whisker plots, summarizing all the data for "10% portions". This adds median and percentiles for all MEs, depending of the continental fraction.

The authors did not analyze the variability of kappa with respect to different air mass origins due to high uncertainty. However, the authors can present some representative kappa values for CEs only and MEs only. They may not show a meaningful difference but such result would support the argument that large particle concentration did not have significant dependence on air mass origin.

Thanks for suggesting to add this (we understand that with "large particle concentration" you meant the "concentration of large particles", as otherwise we would not understand your comment). There is the following new text now on page 15, line 18:

"Separate analysis of  $\kappa$  for CEs and MEs results in 0.99 +/- 0.18 (to which ~ 64% of all separate  $\kappa$  values contribute) and 1.05 +/- 0.20, respectively. There is no clear difference in hygroscopicity of the here analyzed large particles of roughly 110 nm, independent of the time the air mass had been over the continent. This points towards common sources of these large particles for both, CEs and MEs, which are discussed in the following."

P5, L16: It was mentioned that necessary corrections were very high in the two LAS channels below 100 nm when the authors used ammonium sulfate particles. What about 100 nm particles? I also wonder if

the differences between LAS and SMPS were similar for ammonium sulfate particles and PSL particles of the same selected size of 100 nm.

The information you ask for was partially given in the first version of the manuscript ("Necessary corrections were high in the two LAS channels below 100nm (around +70%), distinct in the two channels around 100 nm (+ 10%) and low in the other size ranges up to 800nm (between 1 to 5%, negative and positive corrections).") We changed the wording of this sentence to:

"... below 100nm (around +70%), +10% in the two channels around 100 nm and low in the other size ranges up to 800nm ..."

For calibration at the largest sizes (700 nm and 800 nm), concentrations of ammonium sulfate particles were much lower than those of the PSL particles (for the latter we got 40 cm-3 and 20 cm-3 at 700 nm and 800 nm, respectively), so here only the PSL signals were used for correction. We therefore confined the size range we give for the calibration with ammonium sulfate particles in the text, excluding 700 nm and 800 nm. At 500 nm, particles from both ammonium sulfate and PSL indicated that only a small correction was needed (3% and -0.2%, respectively). Here the larger value was chosen for the correction. At 100 nm, the PSL signal was rather broad and difficult to evaluate, so here the ammonium sulfate signal was used for the correction. We added the following to the text now:

page 6, line 19: "At 500 nm, both ammonium sulfate and PSL particles resulted in similar signals, however, at 100 nm signals for PSL appeared in a broad range of channels, so here only ammonium sulfate particles were used for the validation."

The estimated kappa values were compared with comparable measurements in some other studies. The same should be done for total particle concentrations and CCN concentrations to illustrate how these concentrations fair with other measurements in the Antarctic region.

Thank you for this remark, you are correct. We included in Sec. 3.1:

page 11, line 24: "O'Shea et al. (2017) and Kim et al. (2017) both also report  $N_{CCN}$  determined during austral summer, however, at coastal Antarctic locations. O'Shea et al. (2017) show  $N_{CCN}$  of approximately 20 cm-3, 120 cm-3 and 250 cm-3 on average at 0.08%, 0.2% and 0.53% *SS*, respectively, and just under 200 cm-3 at 0.4% *SS* are given in Kim et al. (2017). These  $N_{CCN}$  are roughly 50% above those determined herein, across all supersaturations. This might be explained by our measurement site's larger distance to the Southern Ocean. As we will show below, air masses often traveled over Antarctica for extended times before reaching our measurement station, which might be connected to an increased wash out of CCN by precipitation along the way."

Minor comments:

P2, L24: positive -> increasing

Done.

P5, L29: also -> at; Apparently the authors observed CCN at 1% SS. How often? What does "adjusted" mean here?

These measurements were done to follow recommendations for CCNc operation, i.e., including measurements at 1 % supersaturation to compare  $N_{CN}$  and  $N_{CCN}$  for conditions when all particles should be activated. With "adjusted" in that respect, we meant that 1 % was set as the supersaturation. This was done a few times after setting up the CCNc newly at each beginning of measurements an austral summer. However, it showed that these values could not be used for the planned comparison, as there were high concentrations of particles that are so small that they are not even activated at 1 % supersaturation. The text was changed to:

page 6, line 34: "For consistency checks between  $N_{CN}$  and  $N_{CCN}$ , additional measurements at a supersaturation of 1 % were made a few times during each season. Respective values for  $N_{CCN}/N_{CN}$  generally were between 0.8 and 0.9. As we will discuss later, the aerosol at PE station is strongly dominated by particles in the nucleation- and Aitken-mode size range, and at 1% SS, not all particles were activated during times when the supersaturation was set to 1% (for example, activation of particles will occur down to 36 nm and 24 nm for an hygroscopicity parameter  $\kappa$  of 0.3 and 1, respectively). Hence this consistency check could not be applied here."

And we added (with respect to the calibrations at TROPOS before the austral seasons):

"Besides for calibration curves for the CCNc, also  $N_{CCN}/N_{CN}$  were derived for particles of different sizes between 120 nm and 200 nm at SS between 0.2% and 0.7%. On average  $N_{CCN}/N_{CN}$  were 1.01, 0.99 and 0.96 for the three different years."

P6, L1-L2: It does not read smoothly. Rewrite this part.

Done.

P7, L27: remove one 'influence.'

Done.

P8, L8: 88152 is not a multiple of 15. Why?

This following explanation was similarly added to the manuscript: "(note that a few trajectories were excluded from the analysis as they could not be properly calculated due to problems in the input data)"

P8, L9: here and at other places, the words "trajectory segment endpoints" could be misleading. In fact, the endpoint is the starting point of the air mass movement. Hopefully the authors can suggest better words.

We changed all occurrences to "trajectory points".

P9, L20: panel C Figure 6, 7 -> panel C of Figures 6, 7

Done.

P12, L34: a kappa value of e.g., kappa = 0.61 -> a kappa value of 0.61

Done.

P13, L5: I am not sure what you mean by "in the same order than the variability of the value itself." Rewrite it.

We added:", i.e., the uncertainty in the derived  $\kappa$  values can be explained based on measurement uncertainties."

P13, L14: during which -> when

Done.

P13, L21: "ammonium sulfate" can be replaced by "sodium chloride" if kappa is 1.75.

The citation here from Asmi et al. (2010) referred to a growth factor (not  $\kappa$ ). This sentence was slightly extended in that we now give growth factors from that citation for three different particle sizes together with the growth factors reported for ammonium sulfate in this publication, which are similar:

page 16, line 27: "They also found the Antarctic aerosol particles to be very hygroscopic with an average hygroscopic growth factor of 1.63, 1.67 and 1.75 for 25, 50 and 90nm particles, respectively, at 90% RH, which is similar to the hygroscopic growth factor of ammonium sulfate particles at 90% RH (given as 1.64, 1.68 and 1.71 for these three different sizes in Asmi et al., 2010)."

P14, L34: the here presented data -> the data presented here

Done.

Figure 4: Explain the unit "g s/m3."

The units are based on the release of a known quantity of particles in g over an integrated time period and the results are displayed per grid box which has a volume component as it includes the elevation of the grid box. Hence gs/m3. We feel that this is not needed to be explained in detail in the manuscript, as it is very model-specific.

Figure 6: The legend of color scale in the bottom panel should be dN/dlogD instead of dN.

Done.

**CCN measurements at the Princess Elisabeth Antarctica Research Station during three austral summers**

Paul Herenz1, Heike Wex1, Alexander Mangold2, Quentin Laffineur2, Irina V. Gorodetskaya3,4, Zoë L. Fleming5, Marios Panagi5, and Frank Stratmann1

1Leibniz Institute for Tropospheric Research, Leipzig, Germany

2Royal Meteorological Institute of Belgium, Brussels, Belgium

3Centre for Environmental and Marine Studies, Department of Physics, University of Aveiro, Aveiro, Portugal

4Department of Earth and Environmental Sciences, KU Leuven, Belgium

5National Centre for Atmospheric Science, Department of Chemistry, University of Leicester, Leicester, UK

Correspondence to: Heike Wex (wex@tropos.de)

**Abstract.**

For three austral summer seasons (2013-2016, each from December to February) aerosol particles arriving at the Belgian Antarctic research station Princess Elisabeth (PE), in Dronning Maud Land in East Antarctica were characterized. in terms

- 5 of This included number concentrations of total aerosol particles  $(N_{\rm CN})$  and cloud condensation nuclei  $(N_{\rm CCN})$ , the particle number size distribution (PNSD), the aerosol particle hygroscopicity and the influence of the air mass origin on  $N_{\rm CN}$  and  $N_{\rm CCN}$ . In general  $N_{\rm CN}$  was found to range from 40 to 6700 cm-3 with a median of 333 cm-3, while  $N_{\rm 
[revised manuscript text omitted]

---

## Author Comment (AC2) · 24 Aug 2018

Review of "CCN measurements at the Princess Elisabeth Antarctica Research Station during three austral summers" by Paul Herenz et al.

We thank Reviewer 2 for the revision of our manuscript (including the comments in the .pdf). Below you can find our answers to your comments and suggestions in blue print. We added numbers (in blue print) to separate parts of your review for easier reference within this document. New text is indicated by quotation marks and printed in italic. When we refer to text from the originally submitted version of the manuscript, the text is in quotation marks and printed in green.

In general, we were somewhat astounded concerning some of the criticisms voiced here, as the reviewer seems to have misunderstood our interpretations a number of times. We may not have emphasized certain points enough and now try to express ourselves more clearly. However, some referee comments refer to statements we did not make or point out missing facts that had, however, already been included in the original version. We discuss all of this in detail below.

This study describes the aerosol particle number concentration (CN) and cloud condensation nuclei number concentrations (N_CCN) at various supersaturations at the East Antarctic research station Princess Elisabeth. Also, limited particle number size distribution data, aethalometer and meteorological data are available. The authors focus on the austral summer measurements between 2013 and 2016. They analyze CN and CCN data additionally with two methods: (a) a particle dispersion model to quantify the fractional contribution of air masses from certain source regions that the authors define, and (b) with a Hysplit based potential source contribution function model to illustrate the potential source regions for the highest 25 % occurrences of CN and N_CCN. The study finds that N_CCN can vary by 2 orders of magnitude reaching up to 1300 cm-3 for supersaturations (SS) between 0.1 and 0.7%. The Aitken mode is the dominant particle mode and 94 % of the particles are < 90 nm. Most air masses reside relatively long over the ice sheet before reaching the station, while a minor fraction arrives more directly from the Southern Ocean. The latter types of air masses bring higher CN and N_CCN concentrations.

1) The contributions to the hygroscopicity parameter kappa, which ranges widely, are discussed, but no conclusion is derived.

We are not sure why the reviewer says that no conclusion is derived concerning the hygroscopicity parameter $\kappa$. Concerning particle hygroscopicity, discussed in Sec. 3.3, the original manuscript already started explaining that "the hygroscopicity parameter $\kappa$ can only be inferred for SS=0.1%, for which the median $d_{crit}$ was determined to be 110 nm. For higher SS [...] $d_{crit}$ is below the lower size limit of the measured PNSD." This was and still is followed by a discussion of roughly one page in length on the connection of a range of $\kappa$ values to chemical composition, followed by other results obtained in Antarctica and their connection to possible particle sources, which resulted in: "Summarizing, we conclude that the few large aerosol particles we observe for sizes of and above ≈ 110 nm may partially originate from NPF and subsequent growth. However the majority of these aerosol particles likely consist of sea spray particles or even more so of sea salt particles emitted over sea ice regions."

Nevertheless, some text was now added to this latter part due to another request further down (see at 7). Altogether, a few additions were made in Sec. 3.3, and instead of pasting a whole page here, please check the (highlighted) changes in the version of the manuscript attached at the end of this answer (in this version of the manuscript, deleted text is given in yellow while newly added text is given in blue).

There is no doubt that this data set needs to be published, because it furthers our understanding of CN and N_CCN in eastern Antarctica, a region that is heavily undersampled. The manuscript is clearly structured and generally well written. Before the manuscript can be published, however, it will need to undergo major revisions which are pointed out in the following with subsequent more specific comments. Technical and some specific edits can be found in the attachment.

2) First, it is unclear why the authors use the NAME dispersion model to create the footprints with the regional classification and the PSCF based on Hysplit for the potential source contributions. These are two different models used for similar purposes. There is no discussion in how far the models produce similar or deviating results, especially in light of the presumably different meteorology (GDAS vs UM meteorological field data). Either, option a), there needs to be a very clear explanation why two different models are used, including a comparison between them to show that the results are comparable, or, option b), either the footprint analysis needs to be done with Hysplit or the PSCF needs to be done with NAME. This might require slight modifications of how the models are run.

Both models were used in the set-up in which they are typically used, as our goal was to see if different models yield similar results. In this context it would not make sense to use the same back-trajectory analysis for both methods, as then one necessarily would be used differently from how it typically is done, and the same issues of the meteorology (if there were some) would influence both models. We agree that this was not communicated clearly, therefore we added to the text:

page8, line 20: "*As will be described in more detail below, the two models are based on different sets of back-trajectories, i.e., they were used in the framework in which they have been tested and applied in the past.*"

Still, it showed that both models have their particular strengths, so somewhat complementary information could be obtained from them, as can be seen when comparing Sec. 3.1 (discussing results from the NAME model) and Sec. 3.2 (discussing the PSCF). However, both models agree in the following: High particle concentrations were only observed during MEs (Marine Events, i.e., when air masses spent more than 10% of the time over the Southern Ocean and / or Productive Zone) by the NAME model. Similarly, the PSCF indicated that regions contributing with a high potential to the highest observed particle concentrations were the Southern Ocean (and the Productive Zone for particles in the size range of ~110nm). This was mentioned in the original manuscript in Sec. 3.2 (where PSCF results are described) and has now been extended:

page 14, line 18: "Hence, the Southern Ocean is likely to be the dominant source region leading to an enhancement in $N_{CN}$ measured at PE, while the Antarctic continent itself is not likely to act as a particle source. This is in accordance with results discussed in Section 3.1, i.e., the low variability of measured number concentrations during CEs *and the occurrence of high values of $N_{CN}$ observed for air masses connected to MEs*."

Second, the regional classification does not make sense and is misleading. The authors imply that the "Reactive Zone" is the largest source of particles and that ammonium and organic nitrogen might be an important contribution to particle mass. Both aspects are likely wrong:

We did not intend to describe the Productive Zone (formerly called Reactive Zone) as the largest particle source, and do not think that we did. The Productive Zone was typically mentioned in conjunction with the Southern Ocean in the original manuscript, e.g., when summarizing both in Marine Events (MEs). Also DMS (as a gaseous precursor) and sea spray emissions ("primary particles") had been explicitly mentioned as contributors to sources for particles we observe. Our aim was (and is) to determine areas from which air masses with high particle concentrations may come from, and both NAME and PSFC analysis indicate the Southern Ocean and Productive Zone as such areas. It is not wrong to assume that all kinds of precursor gases (DMS but also ammonium and organic nitrogen) that are known to be emitted in these regions MAY contribute, which is what we did. Below we added some exemplary passages from the original manuscript to show that the above statement made by the reviewer, together with some others following below, are not correct:

original manuscript, page 10, line 13: "Vice versa, during 39% of the time the proportion of the Reactive Zone plus the Southern Ocean region was larger than 10%, which we from now on call Marine Events (ME)."

original manuscript, page 10, line20: "But also the largest values for $N_{CN}$ and $N_{CCN}$ are only observed during MEs, as the Reactive Zone and the Southern Ocean region potentially represent source regions for primary and secondary formed aerosol particles."

(In this regard, we now clarified the following in the text, in case that this might have led to misunderstandings:"… potential to emit either primary particles *(i.e., particles from sea spray in this case)* or precursors for secondarily formed particles *(i.e., for NPF)*". )

original manuscript, page 7, line 17, concerning the definition of the Productive Zone: "Chlorophyll can be used as a proxy for dimethyl sulfide (DMS) (Vallina et al., 2006), which in turn plays a role in new particle formation (Liss and Lovelock, 2008)."

original manuscript, page 9, line 28: "That is indicative for a high amount of freshly, secondarily formed aerosol particles, which form from precursor gases emitted from the Reactive Zone as e.g., ammonia and DMS."  (This latter sentence now also explicitly mentions the Southern Ocean as a source for these precursor gases, see our answer to your comment concerning original page 9, line 25.)

Also, concerning the Productive Zone, we did mention sea spray production as a possible source explicitly (i.e., "primary particles"):

original manuscript, page 7, line 5: "The Reactive Zone includes the following regions that are known to have the potential to emit either primary particles or precursors for secondarily formed particles."

a) The larger source of particles will be the open ocean through sea spray production (Quinn et al., 2017), because this area is much larger than the "Reactive Zone". Generally the particles are acidic and not neutralized from ammonia emissions. Those have a limited regional effect.

We carefully went through the text and added the Southern Ocean and sea spray explicitly where they might have been missing, although, as shown above, they had been mentioned before already:

Abstract: "For *these* particles *of* ≈ 110 nm *the Southern Ocean together with parts of* the Antarctic ice shelf regions were found to be a potential source region. *While the former may contribute sea spray particles directly*, … "

page 9, line 2: "… *the Southern Ocean may contribute sea spray particles (in a mode with sizes from roughly 100 nm well up into the supermicron size range) which may make up 20% to 30% of all particles, at least above the ocean (Quinn et al., 2017). But the Southern Ocean also is a source for precursor gases for NPF such as DMS and ammonia (Schmale et al., 2013). These precursors may originate in phytoplankton blooms correlated to increased chlorophyll concentrations and have been described to influence CCN over the Southern Ocean (Vallina et al., 2006; Meskhidze and Nenes, 2006).*"

page 9, line 20: description of the Productive Zone explicitly: "*Sea spray production may occur in this region (Quinn et al., 2017).*"

page 12, line 3: "*That is indicative for a high amount of newly formed aerosol particles, which form from precursor gases emitted from the Southern Ocean and the* Productive Zone as e.g., ammonia and DMS (see Section 2.3). *The corresponding NPF events occurring during the passage of the air masses to the measurement site likely take place in the free troposphere (Fiebig et al., 2014; Quinn et al., 2017).*"

page 16, line 1: "*While the lowest κ values we determined point towards a contribution of sulfate containing particles in the here examined particle size range of around 110 nm,* the median κ of 1 might even point towards a dominance of sea salt. *This agrees with sea spray particles being generally larger in size, compared to particles formed during NPF, so that they might contribute to particles in this size range, and it also agrees with an observation made at the Aboa research station, where sodium chloride was found for larger particles with sizes above 100 nm (Teinila et al., 2000).*"

page 17, line 4: "*Therefore,* the majority of these aerosol particles in *the size range of* ≈ 110 nm likely consist of sea spray particles *originating from the open ocean or* of sea salt particles emitted over sea ice regions, a statement we base on their comparably high κ values.*"

However, we do indeed assume that the majority of all particles (by number, not by particulate mass) comes from new particle formation (NPF) and growth, where, as we said before (see above) sulfate (from DMS) and ammonium and organic nitrogen can be assumed to contribute. NPF and growth has been observed as an important particle source for a number of Antarctic sites in the past, with the NPF possibly occurring in the free troposphere, in accordance with a point you raise further down (see at 4). We had cited some studies on this topic in the original manuscript:

original manuscript, page 9, line 30:" However, several other studies at coastal Antarctic sites report PNSD measurements that show pronounced and dominant Aitken modes during austral summer (e.g., Asmi et al., 2010; O'Shea et al., 2017; Kim et al., 2017)."

There is more literature describing that the largest fraction of the particles observed at different Antarctic stations during austral summer comes from new particle formation, while sodium chloride (related to sea spray particles) was found to occur only for sizes of ≈ 110nm. So based on your criticism, we elaborate on this topic more now in the introduction:

*"In general, there is a yearly trend in particle number concentrations, with maximum values in austral summer (Kim et al., 2017; Fiebig et al., 2014). Fiebig et al. (2014) conclude that these cycles are common across the Antarctic Plateau (including the Troll research station 235km from the Antarctic coast but still*

*2000km away from the South Pole), with free tropospheric air masses contributing to air detected at ground. The highest particle concentrations found in austral summer are frequently reported to be due to NPF events (Asmi et al., 2010; Koponen et al., 2003; Kyrö et al., 2013; Weller et al., 2015, Kim et al., 2017). Particles formed during NPF events are likely related to sulfate and ammonia containing compounds that were found in the particulate phase in the submicron size range (Teinila et al., 2000; Wagenbach et al., 1988; Schmale et al., 2013). Precursor gases for NPF can originate from the Southern Ocean (e.g., DMS (dimethylsulfid), Weller et al., 2015; Schmale et al., 2013) and possibly also from e.g., cyanobacteria in freshwater melt ponds (Kyrö et al., 2013), microbiota from sea ice and the sea ice influenced ocean (Dall'Osto et al., 2017) or the decomposition of excreta from fur seals, seabirds and penguins (Legrand et al., 1998; Schmale et al., 2013).*

*Newly formed particles were sometimes reported to grow to CCN size ranges at the Aboa research station, e.g., in Kyrö et al. (2013) and Koponen et al. (2003) (for the latter only in marine air masses), while Weller et al. (2015) report a maximum size of only 25nm for particles grown from new particle formation for observations at the Neumayer research station. This difference was related to a difference in ground cover at the respective measurement sites, which was ice covered around Neumayer but featured melt ponds around Aboa (Weller et al., 2015).*

*Sodium chloride, supposedly from sea spray, was found for larger particles (well above 100 nm) at Aboa, while the majority of particles was smaller than 100 nm (Teinila et al., 2000). A case of exceptionally high particle hygroscopicity was connected to air masses originating from a region with sea ice and open water at the coastal Antarctic research station Halley (O'Shea et al., 2017).*

*Asmi et al. (2010) assumed that particles and nucleating and condensing vapors from the Southern Ocean contribute to particles observed at Aboa, and observed hygroscopic growth factors for particles of 25, 50 and 90nm that were similar to those they reported for ammonium sulfate."*

Furthermore, the increase in $N_{CCN}$ after an initial drop followed by a strong increase in $N_{CN}$ observed after precipitation events suggests (see original version page 10, line 30 ff and Figure 10), that there are events when particles are growing in size with time, and this can be explained due to aging and the formation of additional particulate mass with time. It is, however, difficult to explain this based on sea spray particles – it cannot be expected that there is a repeatedly occurring source of sea spray particles which produces increasingly large particles over time.

Summarizing, the results we described before, which we hope to describe even clearer now, are: While sea spray likely does contribute to particles observed at the Princess Elisabeth (PE) station (as we said and are still saying, those with sizes of ≈ 110nm and likely above), total number concentrations of sea spray particles can be expected to be rather small, based on what is known from literature and from our observations. This should also be considered that the PE station is neither close to nor on the ocean, but 200 km inland at 1390 m above sea level.

b) The animal colonies are an important source of ammonia and lead to secondary particle mass enhancement. But (a) the source is not that important (Riddick et al., 2016; Riddick et al., 2012) compared to the Southern Ocean sea spray emissions, and (b) the authors do not show whether the ammonia emissions are responsible for the increase in CN and N_CCN at PE because they don't have chemical composition data and the kappa value discussion is too vague. Also, ammonia emissions do not necessarily lead to an increase in particle number concentration. They typically lead to the increase of

particle mass. This has been shown by e.g., Schmale et al. (2013). By the way, it is unclear why the authors use a references from an Arctic study (Croft et al. 2016) when there is a study on ammonia emissions from sea birds and seals in the Southern Ocean (Schmale et al., 2013).

We exchanged the citation, now using Schmale et al. (2013) instead of Croft et al. (2016).

However, as elaborated on above, new particle formation has been shown to be an important source of particle number across Antarctica. New particle formation (i.e., the nucleation process itself) has been shown to need both, sulfuric acid (originating from DMS) and neutralizing substances (Riccobono et al., 2014) which may be contributed from ammonia or organic nitrogen. In that respect, it is impossible to disentangle the contribution of precursor gases for the nucleation event itself or for subsequent particle growth, based on our data-set.

Precursor gases in general may be contributed from the open ocean, but e.g., also from the above mentioned animal excrements, from melt ponds or from the sea when it is still covered with ice. All of these possible sources (besides for the melt ponds) were discussed in the original manuscript in relation to the Productive Zone. The new part of the introduction (see above) summarizes all these sources additionally, now. It is far beyond the scope of this work to disentangle the separate contributions of these possible sources, and we never claimed that we did this. We agree that one source alone will not explain NPF and growth. But as NPF has been observed as the main source for particle number across Antarctica (see the new part of the introduction above), it is reasonable to assume that this applies for PE station, too. Unfortunately, the $\kappa$ values could only be retrieved for comparably large particles (roughly ≈ 110nm), and the majority of these particles seems to consist of sea salt. But this is a different topic, as these large particles only make up a minor fraction of all particles (~ 5%, see Tab. 2).

c) Including the permanently and seasonally covered sea ice areas into the "Reactive Zone" for the reasons given, i.e. organic nitrogen and secondary particle formation, is highly misleading. The contribution of organic nitrogen as found by Dall'Osto et al. 2017 is miniscule. The finding is interesting but the relevance for N_CCN is far from being understood and not large. Also, the authors consider the sea ice zone as static, without taking into account the year to year variability. A proper analysis of the influence of the sea ice region requires using satellite images of the sea ice and marginal ice zone borders for each of the seasons considered in this work. Again, conclusions can be misleading.

What we were and are doing in our work is, to analyze from which regions observed high particle number concentrations may come from. And we find, with both NAME and PSCF, that high particle number concentrations are observed for air masses that have a larger influence from the Southern Ocean and/or the Productive Zone, while low concentrations and low variability are connected to air masses that spent long times over the Antarctic continent. In our discussion, we simply summarized possible sources that may contribute to the observed high particle concentrations. You seem to be of the wrong impression that we tried to assign contributions from different sources to the observations, which is not the case. From our analysis and results we discuss potential sources for precursor gases, no matter how strong they may contribute. There is nothing wrong in mentioning them.

Your suggestion here, "A proper analysis of the influence of the sea ice region" is interesting but not what we were aiming at. This would be a different study.

d) The coastal areas around the continents and islands are another component of the "Reactive Zone". The reason the authors give is the enhanced chlorophyll concentrations. Here again, it does not make sense to statically classify an area as chlorophyll rich throughout the considered seasons. In addition to the fact that blooms will not be active to the same degree throughout the whole summer season and every season and year, there are plenty of other regions in the Southern Ocean where enhanced chlorophyll is typically observed (Valente et al., 2016). Those regions should be included as well. Again, satellite images of chlorophyll-a concentrations for the respective periods would have to be used to come to a trustworthy conclusion.

As elaborated on above, we find a combination of the Southern Ocean together with the Productive Zone as the area where those air masses came from that then showed high total particle number concentrations. In both, the Southern Ocean as well as in the Productive Zone, both sea spray particles and also gaseous precursors of all different kinds can be emitted, as discussed above. You are asking for an extension of our results towards combining high particle number concentrations to areas of phytoplankton blooms, and while this certainly could be an interesting topic in itself, it is far beyond the scope of this study.

e) The term "Reactive Zone" is uninformative. The authors mainly refer to regions with enhanced microbial productivity, hence calling it "productive zones" is more appropriate. Furthermore, in the entire manuscript phytoplankton is not mentioned once. The reason for chl-a detection and VOC emissions that can be converted into particles (e.g. DMS to MSA) is the presence of phytoplankton. This link needs to be included explicitly in the manuscript to describe particle sources in the Southern Ocean and Antarctica.

"Reactive Zone" was changed to "Productive Zone". Phytoplankton was not mentioned explicitly in the original version of the manuscript, however, when defining the distinct areas used for the NAME analysis, chlorophyll, which belongs to phytoplankton and which is thought to cause DMS emissions responsible for new particle formation had been discussed in the original manuscript and described as a source for precursor gases for new particle formation:

"- The marine area up to 200 km from the coasts of the islands and continents (for Antarctic, continent plus ice shelves). These areas are included due to an enhanced chlorophyll concentration in the coastal areas of the Southern Ocean. Chlorophyll can be used as a proxy for dimethyl sulfide (DMS) (Vallina et al., 2006), which in turn plays a role in new particle formation (Liss and Lovelock, 2008)."

We now added:

page 9, line 3: "… the Southern Ocean may contribute sea spray particles (in a mode with sizes from roughly 100 nm well up into the supermicron size range) which may make up 20% to 30% of all particles, at least above the ocean (Quinn et al., 2017). But the Southern Ocean also is a source for precursor gases for NPF such as DMS and ammonia (Schmale et al., 2013). These precursors may originate in phytoplankton blooms correlated to increased chlorophyll concentrations and have been described to influence CCN over the Southern Ocean (Vallina et al., 2006; Meskhidze and Nenes, 2006)."

page 9, line 25: "The comparably coarse division into the different regions used in this study was thought to yield a general idea on the possible origin of particles or particle precursors. A more detailed

*investigation of, for example, the variability of the ice cover or the existence of phytoplankton blooms in the examined regions is beyond the scope of our study."*

Specific comments:

3) PSCF: In how far is the altitude of air trajectories included in the PSCF? An air mass that travels aloft, will likely not be influenced by surface emissions, hence the PSCF result can be misleading if any trajectory altitude is considered. This requires clarification and potentially recalculation. Also, on p. 8, l. 24 the authors state that the criterial are empirical without clarifying where the empirical evidences comes from.

Indeed, you are correct that we did not cut off trajectories above a certain altitude. But nevertheless, the results of the PSCF show us regions with a potential to contribute to high number concentrations of CN and CCN. As transport of precursor gases can also occur in the free troposphere, where then NPF occurs (as you quite correctly indicated, referring to Quinn et al., 2017 below), a restriction of the trajectories to lower levels is not necessary. In fact, this is related to your next comment, where you explicitly ask us to include NPF in the troposphere.

Concerning empirical evidence, the wording was awkwardly chosen, and we clarified:

page 11, line 3: *"While the precipitation filter criteria described here were particularly contrived for our study, we used the weighting function given in Waked et al. (2014), as already stated above."*

4) NAME and PSCF: It seems that mostly surface sources of CN and N_CCN have been considered. Over the Southern Ocean and Antarctica it is assumed that a significant fraction of particles comes from particle formation in the free troposphere e.g., in the outflow of clouds (Quinn et al., 2017 and references therein). This is not considered at all, but needs to be included into a comprehensive discussion of particle sources.

If particles originate from precursor gases that originate from the surface, then ultimately particles will originate from these surface sources of the precursor gases. In that sense, we were somewhat irritated when we read this comment, as NPF is the major source for small particles we discussed throughout the manuscript., e.g., in the original version:

original abstract: "It is shown that the aerosol is Aitken mode dominated and is characterized by a significant amount of freshly, secondarily formed aerosol particles …"

original page 5, line 5: "The Reactive Zone includes the following regions that are known to have the potential to emit either primary particles or precursors for secondarily formed particles:" The list following this previous sentence lists possible sources for gaseous precursors, including DMS as well as ammonia and organic nitrogen as precursors for NPF.

original page 9, line 25: " … 36% the aerosol particles are smaller than roughly 35 nm. That is indicative for a high amount of freshly, secondarily formed aerosol particles, which form from precursor gases emitted from the Reactive Zone as e.g., ammonia and DMS …" (We added/adjusted this to: "… *emitted from the Southern Ocean and the Productive Zone* …", now to be found on page 12, line 4)

original summary and conclusions: "From this we conclude, that an Aitken mode dominated aerosol prevailed, that includes a significant amount of freshly, secondarily formed aerosol particles."

We did not mention explicitly that these particles likely form in the free troposphere, which is included now. Also Fiebig et al. (2014) found that ground based measurements in Antarctica are influenced by free tropospheric air masses. Both Quinn et al. (2017) and Fiebig et al. (2014) are now explicitly mentioned in this respect in the manuscript.

page 3, line 17: "*Fiebig et al. (2014) conclude that these cycles are common across the Antarctic Plateau (including the Troll research station 235 km from the Antarctic coast but still 2000 km away from the South Pole), with free tropospheric air masses contributing to air detected at ground.*"

page 12, line 5: "*The corresponding NPF events occurring during the passage of the air masses to the measurement site likely take place in free troposphere (Fiebig et al., 2014; Quinn et al., 2017).*"

5) Growth Rate (GR) discussion: On p. 11, the growth rate discussion seems incomplete. There is no connection to the observation discussed in the published literature (Aboa, Dome C) and the observations at PE (no GR was determined at PE). Furthermore, the reader is left puzzled by the statement that the GR was higher at Dome C than at coastal sites. An explanation or at least discussion of this counter-intuitive observation needs to be included.

We extended the discussion on growth rate, page 13, line 22:

"Other studies at Antarctic sites report events of NPF during austral summer, e.g., Asmi et al. (2010) and Weller et al. (2015) at the Finnish research station Aboa and the German Neumayer station, respectively, which are both coastal sites. Järvinen et al. (2013) *even reported the observation of NPF at Dome C, a site in Central Antarctica. Median* growth rates of particles from NPF were ~2.5nm/h at Dome C *throughout the year, and 3.4 nm/h and 0.6 nm/h for particles up to and above 25 nm, respectively, in the austral summer. At the coastal site of Aboa, variable growth rates were reported, ranging from 0.8 nm/h to 2.5 nm/h reported in Asmi et al. (2010) and from 1.8 nm/h to 8.8 nm/h derived in Kyrö et al. (2013), while growth rates were only ≈1 nm/h for the costal site of Neumayer (Weller et al., 2015). While* it was also described that particles rarely grew to sizes larger than ~25nm *at Neumayer (Weller et al., 2015),* i.e., that they do not reach sizes at which they can readily act as CCN, *growth of newly formed particles into the CCN size range was reported for Aboa, likely due to precursor emissions from local meltwater ponds (Kyrö et al., 2013) or due to precursor gases advected to the site with marine/coastal air masses (Koponen et al., 2003). The surprisingly high growth rates observed at Dome C may be related to air masses that had picked up precursor gases for the formation of particulate matter over the Southern Ocean or the region defined as Productive Zone herein, and that were subsequently transported in the free troposphere followed by descent over Antarctica (Fiebig et al., 2014). This likely is a process occurring widely spread in Antarctica, for which not the availability of precursor gases but rather the photooxidative capacity regulates the connected NPF and particulate growth (Fiebig et al., 2014). Tropospheric NPF with subsequent growth therefore likely also explains the above described observations at the PE station.*"

**6)** Kappa: There is no explicit discussion of the sources of uncertainty for the derivation of the hygroscopicity parameter kappa (p. 12, l. 32 f), just the reference to the Monte Carlo method. More details are necessary.

Besides for referring to a work in which this method is explained in detail for a similar data-set (only taken in the Arctic), we added the following:

page 16, line 7: "*In this approach, uncertainties of input parameters needed for the calculation of $\kappa$ are combined, namely the uncertainties for particle sizing and counting as well as for the supersaturation adjusted in the CCNc. During Monte Carlo simulations, these parameters are randomly varied within their uncertainty range during a large number of separate runs (10000 runs in this study) to yield the uncertainty of the derived $\kappa$ based on the uncertainty of the input parameters.* This analysis shows that the uncertainties in our $\kappa$ values are in the same order than the variability of the values itself, *i.e., the uncertainty in the derived $\kappa$ values can be explained based on from measurements uncertainties.*"

**7)** Origin of particles > 110 nm: The authors list 2 options (p. 13, l. 21ff): NPF and subsequent growth, and sea spray / sea salt particles emitted over the sea ice region. Based on the previously cited literature on GR (1 – 2.5 nm/h) a sustained growth over a period of 40 to > 100 hours would be needed which is unlikely given the lack of condensable gases. Why is there no discussion about cloud-processed particles and emissions from the open ocean? Those are two mechanisms likely to produce accumulation mode particles. Those mechanisms should also be considered in the discussion on p. 12, l. 1-2.

We did not mean to say that the sea spray particles would only be emitted over the sea ice region. We restructured the sentence, so that this cannot be mistaken any more. Cloud processing indeed had not been mentioned before, but is mentioned now, page 16, line 35:

"Summarizing, we conclude that the few large aerosol particles we observe for sizes of and above ≈ 110nm may partially originate from NPF and subsequent growth. *In this respect, it should also be explicitly mentioned that cloud processing of particles also adds mass to those particles that are activated to cloud droplets (Ervens et al., 2018, and references therein), potentially aiding the growth of particles formed by NPF into here discussed size range. However, particulate mass added during cloud processing will not have $\kappa$ values above these of sulfates. Therefore*, the majority of these aerosol particles in the size range of ~110nm likely consist of sea spray particles *originating from the open ocean* or of sea salt particles emitted over sea ice regions, *a statement we base on their comparably high $\kappa$ values*. This fits to the results presented for $N_{CCN;0:1\%}$ in Section 3.2, showing the marine areas in coastal proximity and especially the shelf ice regions as potential source regions."

**8)** Size of particles as function of transport time: This argument brought forward on p. 12, l. 2ff is not convincing keeping in mind the comment above. Particle size can be a function of the emission source and the cloud-processing as well. I recommend plotting the particle size vs transport time to back up this statement.

The PSCF results for CCN measured at different supersaturations, which are discussed above this argument (formerly page 12, line 2ff), are exactly what you are asking for, if only for two different particle sizes. We adjusted the text slightly for a better understanding:

"The PSCF of $N_{CN}$-$N_{CCN;0:7\%}$ *(particles with sizes below ~ 35nm)* shows a large area of high signals between 40° W and 60° E. When calculating transport times based on air mass back trajectories, an average transport time of 5.1 days from this area to PE station is obtained. The PSCF of $N_{CCN;0:7\%}$ *(particles with sizes above ~ 35nm)* shows the largest area of high signals in a region between 140° W and 80° W, for which the average transport time to the PE station is 8.8 days."

This means that there are two points in the comparison of particle size vs transport time given in here (and visually in Fig. 11, the figure discussed in this part of the text). As we did not measure particle size distributions below 90 nm, a more detailed analysis of the kind you are asking for is not possible with our data-set.

9) Technical and some more specific suggestions from the .pdf:

The corrections we made based on the reviewer's suggestions in the .pdf of the manuscript are marked in the "tracked changes"-version, besides for smaller ones (e.g., capitalizations, missing "s") which were done but not marked, to ensure the readability of this version. Those comments that need a reply are listed here. All of the other > 60 remarks were simply implemented as suggested by the reviewer but are not separately listed here.

original manuscript, page 1, line 11: The fraction of time during which CEs (continental events) were observed is now included.

original manuscript, page 1, line 14: Our measurements do not occur over a marine area, and in the sentence in question, we refer to the origin of the precursor gases for new particle formation (NPF) ("marine precursors"), not to the NPF event itself. We do not say anything about the location where the NPF occurs, specifically, we are not saying that NPF occurs over the ocean in the marine boundary layer directly. A marine origin is what our analysis suggests for the air masses for which we find high number concentrations of particles (see Fig. 11), and that is what we refer to here. The NPF may still have occurred in the free troposphere (as you mention wrt. Quinn et al., 2017, and which is also said in Fiebig et al., 2014). We elaborate on this more later in the text and in our answers to this review (see above). Here, we changed the sentences in question slightly to:

"MEs however cause large fluctuations in $N_{CN}$ and $N_{CCN}$ *with low concentrations likely caused by scavenging* due to precipitation *and high concentrations likely originating from* new particle formation (NPF) based on marine precursors. The application of HYSPLIT back trajectories in form of the potential source contribution function (PSCF) analysis indicate, that the region of the Southern Ocean is a potential source of precursor gases for Aitken mode particles.*"

original manuscript, page 3, line 1-2: There was nothing wrong with how it was originally formulated, therefore we prefer to leave this part it as it was.

original page 3, line 3-4: A number of references was inserted, together with a whole new paragraph (see above at 2a)).

original manuscript, page 4, line 11: Not much would be gained by showing three separate (and quite similar) wind roses, so we prefer not to add them. These data are not used in the analysis anyway and were only thought to give the reader a sense of the synoptic situation.

original manuscript, page 4, line 12: We used the unit m/s throughout the manuscript and therefore left it as is, here. But to our experience, ACP typically sets the units to its standards during typesetting, anyway, therefore they will determine how it will be used in the end.

original manuscript page 4, line 26: Concerning what we mean by a "smooth bend": As the inlet of the CPC is located in the bottom of the front of the CPC, the inlet tubing needs to change from vertical to this inlet. This transition is not a 90° bend, but the 0.7m flexible conductive tubing after the stainless steel tubing crossed roughly 0.5m height to the inlet of the CPC at around 0.5m distance, representing a smooth transition from vertical to the horizontal frontal inlet of the CPC. We changed the text from with only a smooth bend just before the inlet at the front of the CPC to "*in a smooth bend from the ceiling of the container down to the CPC.*"

original manuscript, page 5, line 18: The recommendation by Gysel and Stratmann includes 1 % supersaturation to compare the CCN with the total CN, assuming that most particles will activate. Why did the authors not follow this recommendation and how were CCNC and CPC data compared?

CCNc measurements at 1% SS were done (see original text page 5, line 20). At the same time, the CCNc was calibrated each time before it went to Antarctica (see original text, page 5, line 22). Housing the world calibration center for aerosols at TROPOS, which also includes CCNc calibrations, the instrument has been (and still is) well calibrated. This includes the check of the count rates, compared to equally well calibrated CPCs, and the CCNc that went to Antarctica never had a problem with this, meaning that both agree well. $N_{CCN}/N_{CN}$ at supersaturations for which all calibration particles (ammonium sulfate) were activated were 1.01, 0.99 and 0.96 for the respective calibrations in the different years. However, $N_{CCN}/N_{CN}$ was observed to vary between 0.8 and 0.9. For $\kappa$ of 0.3, 0.7 and 1, at 1% supersaturation, particles down to 36, 28 and 24 nm are activated, respectively, i.e., the supersaturation of 1% can be expected to not be large enough to activate all particles that are present in the Nucleation- and Aitken-mode dominated aerosol we encountered at PE station. We edited the text as follows:

page 6, line 33: "*For consistency checks between $N_{CN}$ and $N_{CCN}$, additional measurements at a supersaturation of 1 % were made a few times during each season. Respective values for $N_{CCN}/N_{CN}$ generally were between 0.8 and 0.9. As we will discuss later, the aerosol at PE station is strongly dominated by particles in the nucleation- and Aitken-mode size range, and at 1% SS, not all particles were activated during times when the supersaturation was set to 1% (for example, activation of particles will occur down to 36 nm and 24 nm for an hygroscopicity parameter $\kappa$ of 0.3 and 1, respectively). Hence this consistency check could not be applied here.*

And we added:

"*Besides for calibration curves for the CCNc, also $N_{CCN}/N_{CN}$ were derived for particles of different sizes between 120 nm and 200 nm at SS between 0.2% and 0.7%. On average $N_{CCN}/N_{CN}$ were 1.01, 0.99 and 0.96 for the three different years.*"

original manuscript, page 5, line 28 and page 6, line 7: It is true that it is mentioned thrice that the station is a zero-emission station. However, this is repeated here to explain why we found what we describe here ("the container was most often exposed to non-contaminated air" for the first occurrence you suggested to delete, and "no relationship between wind speed or wind direction with elevated values for NCN or light-absorbing aerosol" for the second occurrence you suggested to delete), and we feel that it is important to have this information in the context of the sentences.

original manuscript page 6, line 4-5: Information on the aethalometer is now included in the measurement section as follows, page 7, line 11:

"*In addition, also data from an aethalometer (Magee Sci. AE31, 7-wavelength aethalometer) was used. The aethalometer was operated with an inlet flow of 5.5 l/min and, similar as for the other instruments, the tubing through which it was fed was 2 m of flexible conductive tubing, including the inlet on the roof of the measurement container. The measurement interval was set to 60 min. Aethalometer data were analyzed following the guidelines in WMO (2016).*"

original manuscript page 6, line 25: We added "*which represents the probability of a specific location to contribute to high measured receptor concentrations.*"

original manuscript, page 9, line 24: Kappa-Köhler theory was cited when it was first mentioned on page 6, line 11, so a reference is not needed again here.

original manuscript, page 12, line 29: In the original manuscript, this sentence together with the previous one are: "Secondarily formed aerosol particles of marine origin are a result of DMS oxidation and further reactions. They can be expected to contain sulfates, and Petters & Kreidenweis (2007) give a $\kappa$ value of 0.61 for ammonium sulfate". As said, the source of ammonium sulfate is thought to be DMS together with precursor gases containing ammonium and organic nitrogen, forming particulate mass during NPF or particle growth, processes which are more often mentioned in the new version of the manuscript (see our answers to your remarks above) but had also been mentioned in the original version of the manuscript, and which, as we show in references to literature, are well known to occur across Antarctica.

original manuscript, page 12, line 31: As said directly above, ammonium sulfate or other sulfate containing components can reasonably be expected to play a role, which explains the use of the word "even" here. However, we agree that for the here discussed larger particles, this does not play the decisive role. We added, page 16, line 1:

"*While the lowest $\kappa$ values we determined point towards a contribution of sulfate containing particles in the here examined particle size range of around 110 nm,* " and "*This agrees with sea spray particles being generally larger in size, compared to particles formed during NPF, so that they might contribute to particles in this size range, and it also agrees with an observation made at the Aboa research station, where sodium chloride was found for larger particles with sizes above 100 nm (Teinila et al., 2000).*"

original manuscript, page 13, line 16: We checked again the data in Asmi et al. (2010), the work we discuss here, and the growth factors they report are really similar to ammonium sulfate. Hence it makes no sense to also discuss growth factors of sodium chloride here, as this was (and still is) already introduced in the first paragraph of this section (3.3). We made the similarity to growth factors of ammonium sulfate clearer in the respective sentence:

"They also found the Antarctic aerosol particles to be very hygroscopic with an average hygroscopic growth factor of *1.63, 1.67 and* 1.75 for *25, 50 and* 90nm particles*, respectively,* at 90% RH, which is *similar to* the hygroscopic growth factor of ammonium sulfate *particles at 90% RH (given as 1.64, 1.68 and 1.71 for these three different sizes in Asmi et al., 2010).*"

original manuscript, page 13, line 19: We agree that this is a very small kappa value, but as we are citing a publication here, it is just what it is. We even already did (and still do) say in the text prior and following your remark: "Unlike these studies and our findings, Kim et al. (2017) report a lower particle

hygroscopicity. Their results are based on CCN and PNSD measurements that were conducted at the King Sejong Station in the Antarctic Peninsula between 2009 and 2015. For CCN measurements at a SS of 0.4% they found an annual mean κ value of 0.15 ± 0.05, which, however, is the only time such low κ values were reported for Antarctica." As kappa values for Antarctica in general are scarce, we nevertheless prefer to cite this work for an overview of what has been found concerning this topic.

original manuscript, page 14, line 15: The change in wording you suggest here would slightly shift the meaning of the sentence and we meant to say what we said (i.e.,: "Antarctic continent itself was found to not act as a significant source for aerosol particles and CCN." instead of "Antarctic continent itself was not found to act as a significant source for aerosol particles and CCN.") We prefer to leave it as is.

Figure 1: It had been said in the caption of the original version that the view is from ESE, indicating the direction, as you wished for. We now write "*east-south-east*" to make it clearer. The construction that was asked about is a thin mast with ropes attached to the ground onto which Tibetan prayer/wish flags are fixed (in fact, the original mast and flags were mounted from a Tibetan expedition member during the period when choosing the location of PE station and the flags are now renewed each season). It would not add scientific value to mention this in the final paper version.

Figure 3: The purpose of the figure is unclear. It does not contribute any relevant information.

This figure shows how a footprint generated with the NAME model looks like. As this is not simply a group of trajectories, we feel that this information is important for readers who never saw such a footprint, to makes it easier to understand what the model is doing.

Figure 6: We discussed here for some time on how to improve the visibility of the picture, also including colleagues who are typically critical and make good plots. Generally, the color scaling for the CCN data was considered a good one, particularly as there are too many overlapping CCN data to see details, anyway, independent of the plot color we choose. We did, however, reduce the size of the black and red dots, which partially overlay the CCN data, to increase the visibility of the latter. Important numbers to be taken from this plot are summarized in Table 2, and the data will be made available on the database Pangeae if this manuscript is accepted, so that readers interested in more detail can look at the data directly. We hope you are satisfied with our solution, and in case that not, we ask you to make a suggestion on how exactly to change the colors.

Figure 9: We added a second panel to this figure displaying median and percentiles.

Figure 12: It had been said in the original version of the manuscript, that κ values could only be determined for particles with a size of roughly 110 nm. This was first mentioned in the introduction and then again in the first two lines in Section 3.3 on hygroscopicity:

original manuscript page 12, line 14: "For the data set presented here, the hygroscopicity parameter κ can only be inferred for SS=0.1%, for which the median $d_{crit}$ was determined to be 110 nm.").

Following this sentence, it was then referred to Fig. 12:

original manuscript page 12, line 17: "All κ values from the three seasons have a median value of 1 and are shown in a histogram in Figure 12."

To make things even clearer, we also added this information to the caption of Fig. 12:

"*derived for a supersaturation of 0.1%, for which the median $d_{crit}$ was determined to be 110nm.*"

[revised manuscript text omitted]

---

## Referee Report (RR1)

**Second Review of manuscript ACP-2018-245**

The revised manuscript accommodated most of my comments but I find that some more things need to be done for more clarity. Once done, the manuscript can be accepted for publication.

Figure 9b shows only the differences of $N_{CN}$. So it should be good to show not only the differences of $N_{CN}$ but also the differences of $N_{CCN}$ at several supersaturation values between CEs and MEs (perhaps excluding precipitation days in MEs) in Table 2. Since difference of aerosol concentration between CEs and MEs is depend on aerosol size, adding representative percentile concentration of CCN would be informative.

Figure 10: $N_{CCN}$ at different supersaturations did not seem to show time lag. Can the authors insist that the nucleated particles grew to a size that could act as CCN as time progressed? Concentration of large particles (SS=0.1%) also tended to increase with time.

Figure 11: Add the PSCF results for $N_{CN > 90\ nm}$ (LAS). I think this can be compared to the result of $N_{CCN,07}$ and $N_{CCN,01}$.

---

## Author Response (AR2)

Dear Editor, der Referees!

We thank you for your efforts. Below are our answers to the points that were raised in the second round of this review process. We hope that we can answer them to your satisfaction. The reviewer comments are given in blue, our answers are given in black. In the updated version of the manuscript, changes are indicated in bold script.

Please let us know if this suffices for you to accept our manuscript for publication in ACP.

 Sincerely,

Heike Wex
* * *
Referee #3

Figure 9b shows only the differences of $N_{CN}$. So it should be good to show not only the differences of $N_{CN}$ but also the differences of $N_{CCN}$ at several supersaturation values between CEs and MEs (perhaps excluding precipitation days in MEs) in Table 2. Since difference of aerosol concentration between CEs and MEs is depend on aerosol size, adding representative percentile concentration of CCN would be informative.

We made a new Figure 9 showing the scatter plots and box and whisker plots additionally for CCN for two supersaturations (0.3% and 0.7%) exemplarily, and adjusted the text accordingly (see page 12, line 15ff). As respective values had been given before (and are still given) in Table 2, no additional new discussion was needed.

Figure 10: $N_{CCN}$ at different supersaturations did not seem to show time lag. Can the authors insist that the nucleated particles grew to a size that could act as CCN as time progressed? Concentration of large particles (SS=0.1%) also tended to increase with time.

Concerning an observed time lag, we now say "accompanied" instead of "followed" (page 13, line 12), as this indeed captures all cases (it's only for some cases that the increase in size in larger particles started delayed).

Concerning the increase in concentrations of large particles, we thank you for noticing that. The absolute changes are quite small, but you are correct. We therefore changed this passage by deleting original lines 12 and 13 on page 13 and changed the remaining text (now page 13, lines 13-14) to:

"In the vast majority these events of increased $N_{CN}$ were accompanied by an increase of $N_{CcN}$ by a factor of at least roughly two at all SS."

Nevertheless, in general, we clearly see newly formed particles, indicated by a strong increase particularly in small particles (~ > 35 nm) and also an increase (often ~doubling) in number concentrations for particles with sizes of roughly 35, 45, 60, 80 and 125 nm (these sizes are rough estimates based on an assumed kappa value of 0.8 and on the supersaturations in the CCNc, and particles of particularly the larger sizes surely can act as CCN). If this increase in concentration at the

larger sizes would not come from newly formed and condensed particulate matter, another explanation would have to be found for the observed increase. The authors are not aware of an alternative useful explanation of the observation. If one is known we are open to discuss it. For the time being, concerning this topic, nothing was changed.

Figure 11: Add the PSCF results for $N_{CN > 90 nm}$ (LAS). I think this can be compared to the result of $N_{CCN,07}$ and $N_{CCN,01}$.

As can be seen in Figs. 6-8 (panel C in all cases), $N_{CN > 90 nm}$ (LAS) and $N_{CCN}$ at 0.1% supersaturation ($N_{CCN,01}$) are very close to each other (symbols in red and darkest blue) throughout the whole time when measurements were made. The PSCF result for $N_{CN > 90 nm}$ (LAS) is therefore very similar to that show in Fig. 11, lower right panel (for $N_{CCN,01}$). Adding it does not add anything new to the work and only unnecessarily makes it longer. Nothing was changed.
* * *
Referee #1

One point I keep wondering about is in how far the PSCF results reflect the source region of precursor gases of secondary aerosol formation. As is now discussed in the work, NPF can occur in the free troposphere, so the location of the formation can be understood as source region. This source region would, however, not reveal where the precursor gases came from because they can travel over long distances in the atmosphere before a change in CN or CCN can be observed (e.g., Woodhouse et al., 2013). It is beyond the scope of this paper to investigate this systematically, but a short mentioning of this would clarify the type of information that can be extracted from the chosen methodological approach. My recommendation is to publish the manuscript with this minor addition.

It is correct that the PSCF does not exactly show where precursor gases came from, NOR does it show where the new particle formation takes place. We added the following (page 14, line 29ff):

"While with this interpretation we assume the source regions to be the regions where NPF may have taken place, gaseous precursors may have been emitted in these regions, too, or may have been transported over some distance. Still, our analysis clearly indicates that the Southern Ocean region is a region potentially acting as a source of the majority of particles observed at PES."

We did, however, not find a way to incorporate the citation you gave. As we understand this cited work, the outcome concerning the Southern Ocean is rather that changes in DMS emissions there will not have a large influence on CCN number concentrations. The statement made above "the precursor gases … can travel over long distances in the atmosphere before a change in CN or CCN can be observed (e.g., Woodhouse et al., 2013)." is, to our understanding, not made in it, unless you refer to what could be envisioned in their Fig. 3b, which, however, is not discussed in their text.

Literature

Woodhouse, M. T., Mann, G. W., Carslaw, K. S., and Boucher, O.: Sensitivity of cloud condensation nuclei to regional changes in dimethyl-sulphide emissions, Atmos. Chem. Phys., 13, 2723-2733, 2013.

---

## Author Response (AR3)

Dear Paul!

Thanks a lot for this thorough check! We followed your recommendations (they are shown here in blue print) as described below. Those to which we do not respond directly we also did not mark in the revised version, as these were all really minor (technical editing). For ease of reading, we rearranged the sequence of your remarks.

Page 15, line 21: Kappa for NaCl should be around 1.5. Petters and Kreidenweis erroneously transcripted it from Koehler et al. (see last paragraph on page 5 and Table 1 in Zieger et al., 2017).

The value given in Zieger et al. (2017) is cited now as well. As Petters and Kreidenweis (2007) also show the discrepancy between kappa determined from hygroscopic growth and activation data, we still kept the citation and added explicitly that these two publications give different values (see bold script on p15, l14-16). As we are not explicitly dealing with NaCl, we would prefer to not elaborate more on this matter.

I'm a bit confused about the exact time frames of the performed measurements. On page 5 (line 26) and in Figure 5 you state that you started measuring in Nov. 2012, while Figures 2 and 6 suggest that you started in Dec. 2013 Please clarify where the missing year is.

The CCNC was operated during the three austral summers from Dec. 2013 until Feb. 2016. This is said in the starting sentence of the abstract, at the end of the introduction (p3, l5-7) and at the end of the first paragraph in the experimental section 2.1 (p4, l30-32, "As the Cloud Condensation Nuclei counter used for this study needs an operator on site, we mainly present data collected from December to February during three subsequent austral summers (2013-2016).").

The CPC has been operated since 2012, and this is what is referred to on page 5, line 26 (beginning of Sect. 2.2: "The CPC was first installed for continuous operation in November 2012.") and in Fig. 5. It was installed at the station all the time and operated (remotely controlled) during the austral winter as long as possible.

To make this clearer, previously to mentioning Fig. 5, we added (p11, l2) the bold text:

"Measurements of $N_{CN}$ throughout the whole year were performed between 2012 and 2016**, i.e., these measurements were done during more extended periods of time than measurements of $N_{CCN}$.**"

Figure 3: Please elaborate a bit more in your figure caption about what is exactly. What are the units? Concentration of what?

We added to the caption: "**It shows the spatial distribution of particles that were released within 3 hours at the PE station and then tracked back in time over 10 days, based on back trajectories. The location of these particles was recorded every 15 minutes in a snapshot and these were then summed up to obtain the footprint.**"

Figure 12: If possible, please improve your y- and x-labels. How many hours of measurements does this correspond to?

We exchanged the labels which were "Number" and "Kappa" by "Counts" and "κ". As said in the caption, this is the data for all 2171 κ values derived for a supersaturation of 0.1% during the three years of measurements. This together with information of the length of the measurements adds up to roughly 600 hours of measurement, an information now added to the caption of the figure.

Page 2, line 16: Maybe add "away" after "far". As "far from" already means "weit entfernt von" (I know you speak German ;-) ), the "away" is not really needed.

Page 3, line 13 and line 24: I would recommend to stick to one tense (past tense as in the other sentences). Done.

Page 4, line 8: To be consistent with the other acronyms, I would us "CCNC" for the cloud condensation nuclei counter instead of "CCNc". Done.

Page 5, line 25 and Page 6, line 6: replace "l/m" by "l/min" Done.

Page 6, second paragraph: Please mention that you mean (optical) diameters for the LAS. Done.

Page 7, line 19: Here and throughout the manuscript: The abbreviation "Sect." should be used when it appears in running text and should be followed by a number unless it comes at the beginning of a sentence (https://www.atmospheric-chemistry-and-physics.net/for_authors/manuscript_preparation.html). Done.

Page 8, line 24: "asl" has been used before and should thus be defined at first occurrence. Done.

Page 11, line 1: Replace "Particle" by "particle". Done.

Page 13, line 10 and 12 (and in Table 2): The unit "cm" should not be in italics (it should be cm$^{-3}$). Done.

Some Latex tips:

- I would not put "PNSD" in math-mode (so remove the $-signs) or would you consider it to be a variable? Done.

- Add '\rm' in the subscript for the critical diameter D$_{\rm crit}$ Done.

Kind regards
Paul.

Kind regards back from Heike & Paul and all co-authors